# Negotiation and honesty in artificial intelligence methods for the board game of Diplomacy

János Kramár[1], Tom Eccles[1], Ian Gemp[1], Andrea Tacchetti[1], Kevin R. McKee ⑩[1], Mateusz Malinowski[1], Thore Graepel[2] & Yoram Bachrach ⑩[1] ✉

The success of human civilization is rooted in our ability to cooperate by communicating and making joint plans. We study how artificial agents may use communication to better cooperate in Diplomacy, a long-standing AI challenge. We propose negotiation algorithms allowing agents to agree on contracts regarding joint plans, and show they outperform agents lacking this ability. For humans, misleading others about our intentions forms a barrier to cooperation. Diplomacy requires reasoning about our opponents' future plans, enabling us to study broken commitments between agents and the conditions for honest cooperation. We find that artificial agents face a similar problem as humans: communities of communicating agents are susceptible to peers who deviate from agreements. To defend against this, we show that the inclination to sanction peers who break contracts dramatically reduces the advantage of such deviators. Hence, sanctioning helps foster mostly truthful communication, despite conditions that initially favor deviations from agreements.

Coordination, cooperation and negotiation play a key role in our everyday lives, from small-scale problems such as safely driving on roads and scheduling meetings to large-scale efforts such as international trade or mediating peace. A key driver to the success of humans as a species is our ability to cooperate with others[1,2]. Systems based on artificial intelligence (AI) control a growing part of our lives, from personal assistants to high-stake decisions such as authorizing loans or automated job market screening[3,4]. Cooperation and negotiation are central to AI[5–11], and AI systems already affect human trade and negotiation through algorithmic trading and bidding[10,12–16]. It is thus imperative that we endow our AI systems with the tools to coordinate and negotiate with others[3,7,11,17,18].

Game playing has been a focus area of AI since its inception. Progress on search, reinforcement learning, and game theory[9,19–21] led to successes in Chess[22], Go[23], Poker[24], control[25], and video games[26,27]. However, the majority of such work deals with two-player games that are fully competitive (zero-sum), which are mathematically easier to analyze[18,28], but cannot capture alliance formation and negotiation.

Similarly, work on games where agent goals are fully aligned[29,30], or two teams of agents engaged in a competition[27,31,32], lacks the need to negotiate. In contrast, many real-life domains require negotiation as the goals of participants only partially align. These domains exhibit tensions between cooperation and competition[7,33–35], making them harder to tackle using AI agents[21,36,37]. Communication has a key role in such settings as it enables us to share beliefs, goals and intents with others, allowing us to negotiate, form alliances and find mutually beneficial agreements[10].

Diplomacy[38] is a prominent 7-player board game that focuses on communication, cooperation and negotiation. An introduction to the rules of Diplomacy is given in Supplementary Note 1. Diplomacy is played on a map of Europe partitioned into provinces, some of which are special and marked as Supply Centers. Each player attempts to own the majority of the supply centers, and controls multiple units (armies or fleets). A unit may support another unit (owned by the same or another player), allowing it to overcome resistance by other units. Due to the inter-dependencies between units, players stand to gain by

[1]DeepMind, 6-8 Handyside Street, London N1C 4UZ, UK. [2]Altos Labs, 2000 Bridge Parkway, Redwood City, CA 94065, USA. ✉e-mail: yorambac@google.com

negotiating and coordinating moves with others. Hence, while ultimately a competitive game, making progress in Diplomacy requires teaming up with others. In each round, every player decides on the actions taken by each of its units, and these moves are executed simultaneously. This yields an enormous action space of $10^{21}$ to $10^{64}$ legal actions per turn, and an immense game tree size of $10^{900}$[39]; for comparison, Chess has fewer than 100 legal actions per turn, with a game tree size of $10^{123}$. The heart of Diplomacy is the negotiation phase occurring before entering moves, where players communicate trying to agree on the moves they are about to execute.

These properties make Diplomacy a key AI challenge domain for negotiation and alliance formation in a large-scale realistic mixed-motive setting. AI approaches for Diplomacy have been researched since the 1980s[40,41]. The standard version is Press Diplomacy, which includes a negotiation phase prior to each move phase. In human games this takes place through conversations in natural language: players may converse privately by stepping into another room, or through a private chat channel when playing online. For AI, researchers have proposed computer-friendly negotiation protocols[42], sometimes called Restricted-Press.

A much simpler version is No-Press Diplomacy, where direct communication between players is not allowed, eliminating the negotiation phase. We contrast No-Press and Press Diplomacy in Fig. 1.

Many AI approaches for Diplomacy were proposed through the years, mostly relying on hand-crafted protocols and rule-based systems[42–46], and falling far behind human performance (both with and without communication). Paquette et al. achieved a breakthrough in game performance[47], training a neural network called DipNet to imitate human behavior based on a dataset of human games. While DipNet has defeated previous state-of-the-art agents by a wide margin, it only handles No-Press Diplomacy, employing no communication or negotiation. Recent work improved the performance of such agents using deep reinforcement learning[39] or deep regret matching[48,49], but still without communication or negotiation. Our Contribution: We leverage Diplomacy as an abstract analog to real-world negotiation, providing methods for AI agents to negotiate and coordinate their moves in complex environments. Using Diplomacy we also investigate the conditions that promote trustworthy communication and teamwork between agents, offering insight into potential risks that emerge from having complex agents that may misrepresent intentions or mislead others regarding their future plans.

We consider No-Press Diplomacy agents trained to imitate human gameplay and improved using reinforcement learning[39], and augment them to play Restricted-Press Diplomacy by endowing them with a communication protocol for negotiating a joint plan of action, formalized in terms of binding contracts. Our algorithms agree on contracts by simulating what might occur under possible agreements, and allow agents to win up to 2.5 times more often than the unaugmented baseline agents that cannot communicate with others. Human cooperation is impeded by the potential impact of breaking agreements or misleading others about future plans[50,51]. We use Diplomacy as a sandbox to study how the ability to break past commitments erodes trust and cooperation, focusing on ways to mitigate this problem and identifying the conditions for honest cooperation. We find that artificial agents face a similar problem as humans regarding breaches of trust: communities of communicating agents are susceptible to peers that may deviate from agreements. When agreements are non-binding, an agent may agree on one course of action in a turn but choose an action violating this agreement in the next turn. We refer to such agents as deviators, and show that even deviators that select their actions using simple algorithms win almost three times as often as peers that never break their contracts. To defend against this, we endow agents with the inclination to retaliate when agreements are violated, and find that this sanctioning behavior dramatically reduces the advantage of deviators. When the majority of the agents sanction peers that break agreements, simple deviators win less frequently than agents that always abide by their contracts.

Finally, we consider how a deviator may optimize its behavior when playing against a population of agents that sanction peers that break agreements, and find that the deviator is best-off adapting its behavior to very rarely break its agreements. Such sanctioning behavior thus helps foster mostly-truthful communication among AI agents, despite conditions that initially favor deviations from agreements. However, sanctioning is not an ironclad defense: the optimized deviator does gain a slight advantage over the sanctioning agents, and sanctioning is costly when peers break agreements, so the population of sanctioning agents is not completely stable under learning.

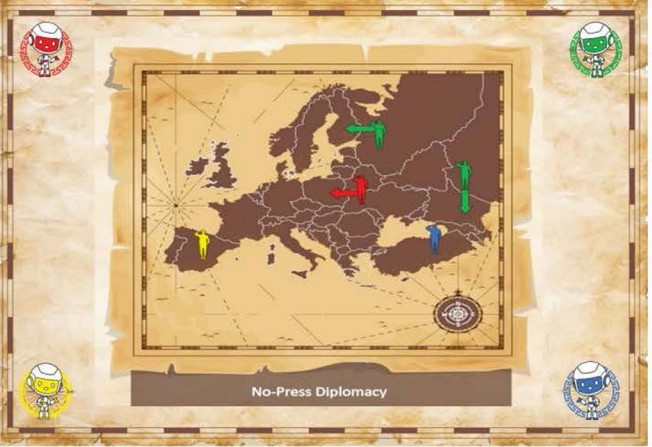
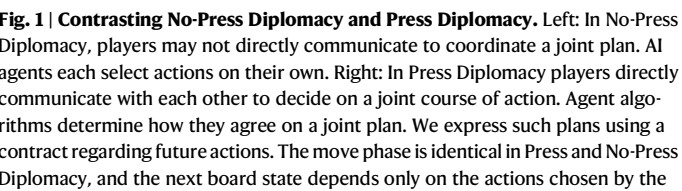
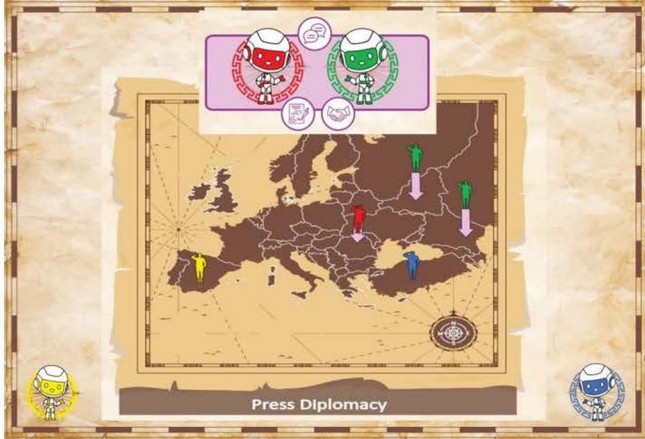

**Fig. 1 | Contrasting No-Press Diplomacy and Press Diplomacy.** Left: In No-Press Diplomacy, players may not directly communicate to coordinate a joint plan. AI agents each select actions on their own. Right: In Press Diplomacy players directly communicate with each other to decide on a joint course of action. Agent algorithms determine how they agree on a joint plan. We express such plans using a contract regarding future actions. The move phase is identical in Press and No-Press Diplomacy, and the next board state depends only on the actions chosen by the players in the move phase. Negotiation in Press Diplomacy only affects how the game progresses if it results in players selecting different actions in the move phase. As players are free to choose any legal action regardless of what they say during the negotiation phase, Press and No-Press Diplomacy are exactly the same game except for the ability of participants to communicate. Background image by rawpixel.com on Freepik.

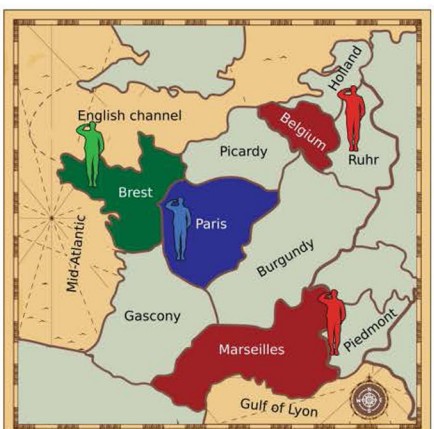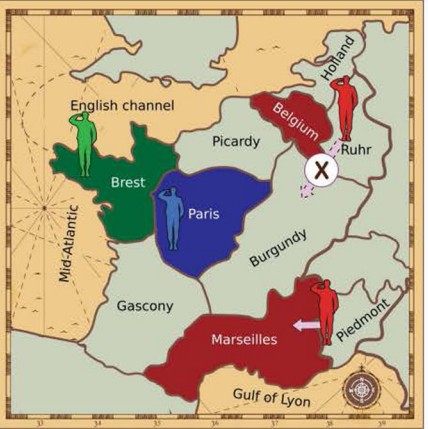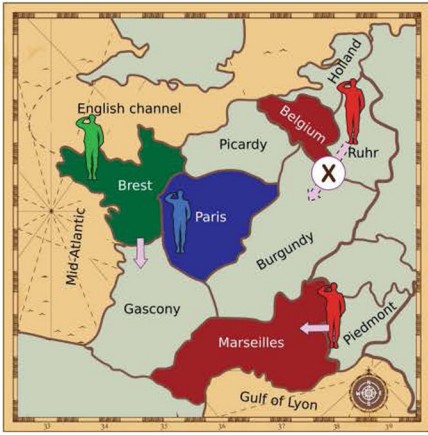

**Fig. 2 | Diplomacy contracts.** Left: a part of the Diplomacy board with three players. Middle: a restriction allowing only certain actions $R_{Red}$ to be taken by the Red player; in this example, the unit in Ruhr may not move to Burgundy, while the unit in Piedmont must move to Marseilles. The set $R_{Red}$ consists only of the actions that fulfill these restrictions. Right: a contract $D = (R_{red}, R_{Green})$ consists of a restriction for both players; in this example, Red's actions are restricted as in the middle, while Green is restricted to actions where the unit in Brest moves to Gascony. Background image by kjpargeter on Freepik.

We view our results as a step towards evolving flexible communication mechanisms in artificial agents, and enabling agents to mix and adapt their strategies to their environment and peers. Our work offers two high-level insights. Firstly, algorithms that reason about the intents of others and employ game-theoretic solutions allow communicating agents to outperform peers through better coordination, even under simple interaction protocols. Secondly, the ability of participants to break from prior agreements is a barrier hindering cooperation between AI agents; however, even in complex environments, simple principles such as negatively responding to the breach of agreements help promote truthful communication and cooperation.

## Results

### Negotiating with binding agreements

Our proposed method takes non-communicating agents trained for No-Press Diplomacy, and augments them with mechanisms for negotiating a joint plan of action with peers. To obtain the initial non-communicating agents we use existing reinforcement learning agents[39], discussed in the section "The sampled best responses procedure". This method constructs neural networks capturing a policy which maps the board state to the game action to be taken, and a value network which predicts the win probabilities of the agents. We provide these agents with a protocol for negotiating bilateral agreements regarding the actions they would take.

We first consider binding agreements, where agents who agree on a joint plan cannot deviate from it later. We view a contract as a restriction over the actions each of the agents may take in the future. For simplicity we focus only on contracts relating to the upcoming timestep. Given two agents $p_i, p_j$ who must act in a state $s$, we denote by $A_i(s), A_j(s)$ the respective sets of actions the agents may take. A potential contract $D$ is a pair $D = (R_i, R_j)$ where $R_i \subseteq A_i$ is the restricted subset of actions each side may take under the contract; an action $a \in A_i \backslash R_i$ is one that violates the contract $D$. One type of contract $D = (R_i, R_j)$ is where $R_i$ consists of a single action: $R_i = \{a_i\}$ where $a_i \in A_i(s)$, meaning that agent $p_i$ commits to taking the action $a_i$ next turn. Two contracts $D^1 = (R_i^1, R_j^1), D^2 = (R_i^2, R_j^2)$ are identical if $R_i^1 = R_i^2$ and $R_j^1 = R_j^2$, denoted as $D^1 = D^2$. Figure 2 illustrates these concepts.

**Protocols.** A protocol is a set of negotiation actions through which agents may communicate and agree on contracts. We consider a set of $n$ agents $P = \{p_1, ..., p_n\}$ about to take simultaneous actions. We consider two general protocols for reaching pairwise agreements regarding future actions. The Mutual Proposal protocol places restrictions on the

actions both sides may take. The Propose-Choose protocol enables both sides to agree on each taking a specific move.

**Mutual Proposal protocol.** Under this protocol every pair of agents $p_i, p_j \in P$ has only a single possible contract between them, depending on the state; we call the specification of this the contract type. Each agent $p_i \in P$ may propose, to each of the remaining agents, to enter into the contract specified by the contract type. We denote the proposal of agent $p_i$ to $p_j$ as $D^{i \to j}$: if $p_i$ does not propose to enter into a contract with $p_j$, this is denoted as $D^{i \to j} = \emptyset$. Two agents agree to a contract if and only if they both propose the contract to one another so $D^{i \to j} = D^{j \to i} \neq \emptyset$. If either side does not make an offer, i.e., $D^{i \to j} = \emptyset$ or $D^{j \to i} = \emptyset$, no agreement is reached, and neither side is restricted in its actions. For Diplomacy we use a Peace contract type. An action by agent $p_i$ violates the peace with agent $p_j$ if one of the $p_i$'s units attempts to move into a province occupied by $p_j$ or enter or hold a supply center owned by $p_j$, or to assist another unit to do so. The Peace Contract $D = (R_i, R_j)$ between $p_i$ and $p_j$ is defined by $R_i$ and $R_j$ containing only the actions that do not violate the peace between $p_i$ and $p_j$.

**Propose-Choose protocol.** This protocol consists of two stages: a Propose phase where each agent may propose a single contract to each other agent, and a Choose phase where each agent selects one of the contracts involving them (one they proposed, or one proposed to them). We refer to contracts proposed in the first phase as contracts On The Table. We denote the contract that $p_i$ proposes to $p_j$ as $D^{i \to j}$. We say a contract $D$ involves agent $p_i$ if $D$ is either a contract $D = D^{i \to j}$ that $p_i$ proposed to another agent $p_j$ or if $D$ is a contract $D = D^{j \to i}$ that some other agent $p_j$ proposed to $p_i$. In the choose phase, each agent $p_i$ may choose only one contract $D$ that involves them out of all the contracts on the table. Denote the contract chosen by $p_i$ as $D_i^*$. In this protocol, two agents only reach an agreement if they choose exactly the same contract, i.e., $D_i^* = D_j^*$ (e.g., both $p_i$ and $p_j$ choose $D^{i \to j}$ or both choose $D^{j \to i}$). In our experiments we enhance the protocol slightly, allowing each player $p_i$ in the Choose phase to either only choose a contract $D^{i \to j}$ or $D^{j \to i}$ for some $p_j$, or to optionally also indicate that they are willing to accept either of $D^{i \to j}, D^{j \to i}$; when both $p_i, p_j$ indicate that they are willing to accept both $D^{i \to j}, D^{j \to i}$ but rank these two contracts in a different order, we select one of the two contracts randomly. We use the Propose-Choose protocol with contracts that completely specify what each unit of each side would do the next turn, i.e., a contract $D = (\{a_i\}, \{a_j\})$. Each agent could potentially propose $n-1$ contracts and could potentially receive $n-1$ contracts, so an agent $p_i$ who wishes to

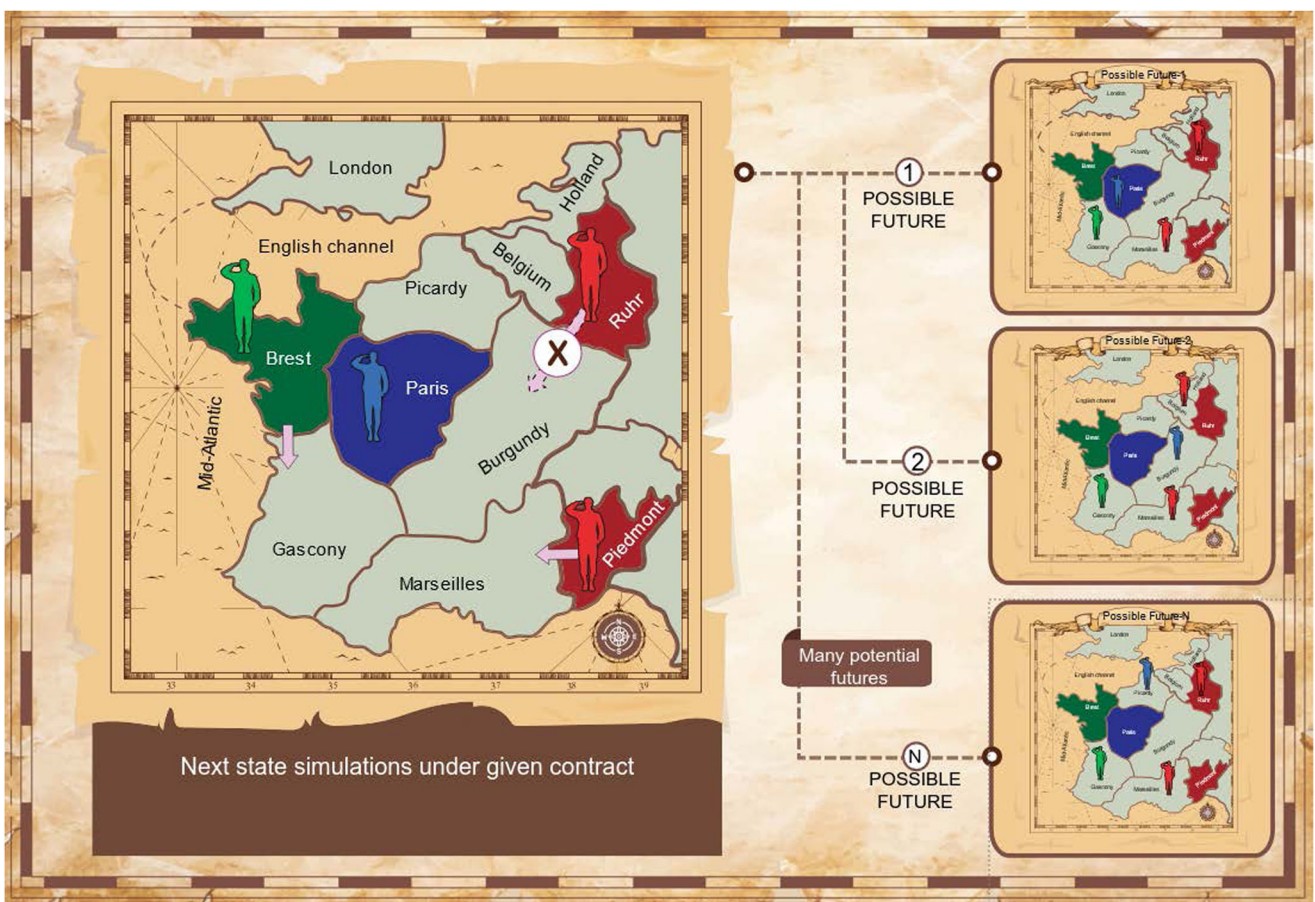

**Fig. 3 | Simulating possible next states given agreed contracts.** Left: current state in a part of the board, and a contract $D = (R_i, R_j)$ agreed between the Red and Green players (same as in Fig. 2). Right: multiple possible next states; the actions of Red and Green are sampled from the restricted policies $\pi^{R_i}, \pi^{R_j}$ allowing only certain actions as specified by the contract, and the actions of the Blue player sampled from the unrestricted policy. Background image by rawpixel.com and kjpargeter on Freepik.

reach an agreement with $p_j$ is competing with the other agents to get $p_j$ to choose their contract. If an agent proposes a contract that is mostly beneficial to itself, the other agent is unlikely to choose that contract. Hence, agents must reason about which contracts others are likely to accept.

**Overview of the Negotiation Algorithms.** We summarize the negotiation algorithms, with full details in the "Methods" section. We propose a method called Restriction Simulation Sampling (RSS) for the Mutual Proposal protocol, and a method called Mutually Beneficial Deal Sampling (MBDS) for Propose-Choose. We evaluate these on Diplomacy using Peace contracts for RSS and contracts fully specifying the next turn moves for MBDS, but our methods generalize to other contract types (see Supplementary Note 6).

Our methods identify mutually beneficial deals by simulating how the game might unfold under various contracts. We use the Nash Bargaining Solution (NBS) from game theory[52] as a principled foundation for identifying high quality agreements. NBS fulfills appealing negotiation and fairness axioms[53] and takes into account the utility of the agents when no deal is reached, called the no-deal baseline or BATNA−Best Alternative To a Negotiated Agreement[54]. Intuitively, NBS strikes a balance between the benefits that either side obtains. We justify the use of the NBS in the "Mutually beneficial deal sampling (MBDS)" section.

Unfortunately, directly calculating expected utilities through game simulation and finding the optimal NBS contract are computationally intractable: the game may unfold in many ways as players may take many possible actions, and there may be a vast space of potential contracts to search through. We address these difficulties through a Monte-Carlo simulation, sampling from the space of potential future states using policy and value functions, which are common machinery in the AI literature[55]. Multiple approaches have been proposed for constructing policy and value functions for Diplomacy, such as imitation learning[47] or regret minimization[48]; we use functions trained via reinforcement learning[39]. Following this training, these functions are held fixed throughout the experiments (colloquially, we freeze the learning). Our methods simulate what might occur in the next turn by sampling from various policies, such as the unconstrained policy of the underlying No-Press Diplomacy agent, or a constrained policy that agent $p_i$ must follow after agreeing to a contract $D = (R_i, R_j)$. This approach is illustrated in Fig. 3.

**Negotiators outperform non-communicating agents**
We show that augmenting agents with our negotiation mechanism allows them to outperform baseline non-communicating agents lacking this mechanism. Both the negotiators and the baseline use the same policy function obtained using reinforcement learning[39], and select actions from it using the same algorithm (the Sampled Best Response procedure[39] described in the "The sampled best responses procedure") section. However, only the negotiators are able to communicate: negotiators interact via the RSS method in the Mutual Proposal protocol, or via MBDS in the Propose-Choose protocol (see the "Baseline Negotiator algorithms" section). Diplomacy has seven players, so we consider $k$ communicating agents and $7 − k$ non-

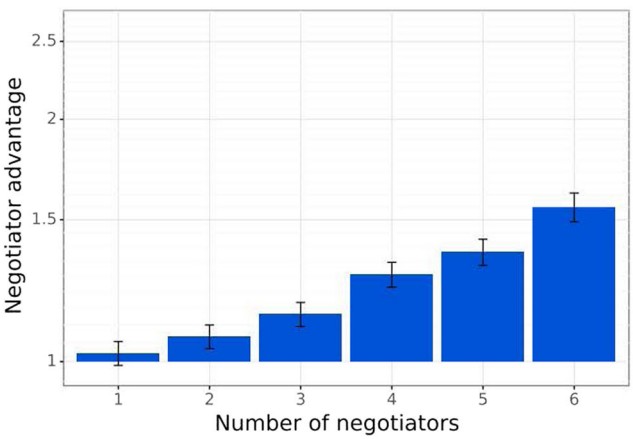 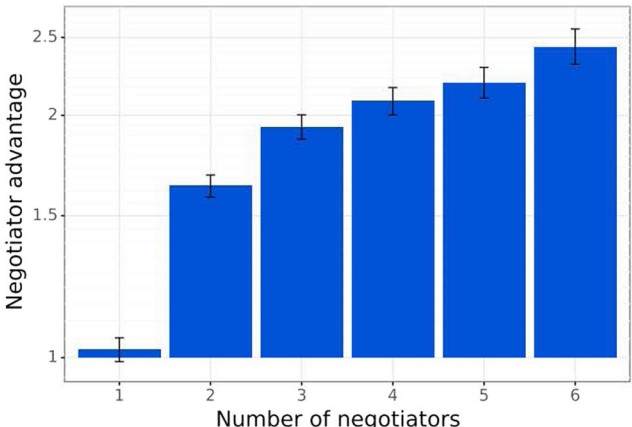

**Fig. 4 | Baseline Negotiators outperform non-communicating agents.** Left: Mutual Proposal protocol. Right: Propose-Choose protocol. The $x$ axis is the number of communicating agents, and the $y$ axis is the winrate advantage of the communicating agent over the baseline, expressed as a ratio $\frac{W_C}{W_N}$, where $W_C$ is the average winrate of the communicating agents and $W_N$ of the non-communicating agents, measured over 10,000 games. The advantage each communicating agent enjoys grows as there are more communicating peers to make agreements with.

The Propose-Choose protocol, where agents agree on an exact fully specified joint action next turn, results in a larger advantage for communicator agents over the non-communicating baseline (though a contract type other than Peace could potentially allow for stronger agent performance in the Mutual-Proposal protocol). Note that for Propose-Choose, the communicating agents exclude the non-communicating agents from consideration to prevent themselves from wastefully choosing contracts that cannot be agreed on.

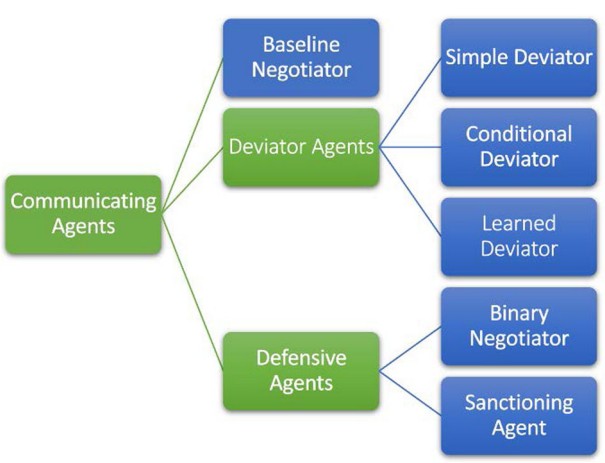

**Fig. 5 | Communicating agents considered in this work.** Each blue block relates to a specific agent algorithm, while each green block relates to a group of agent algorithms.

communicators for $k \in \{1,2,\ldots,6\}$. Figure 4 shows the advantage our communicating agents have over the non-communicating baseline: communicators win up to 2.49 times more often than non-communicators (or 56% more often for the Mutual-Proposal protocol), highlighting the advantage agents gain through cooperation.

### Negotiating with non-binding agreements

A key limitation of our results in the "Negotiating with binding agreements" section is the implicit assumption that agreements are binding—once an agent has agreed to a contract, they do not deviate from it. We now lift this assumption, and consider agents who may agree to a contract in one turn and deviate from it the next. This serves multiple purposes. First, a key feature of the rules of Diplomacy is that agreements made during the negotiation phase are not binding[38] (i.e., communication is cheap talk[56]). Much more importantly, in many real-life settings we can also not assume that agreements are binding—people may agree to act in a certain way, then fail to meet their

commitments later on. To enable cooperation between AI agents, or between agents and humans, we must examine the potential pitfall of agents strategically deviating from agreements, and ways to remedy this problem. Diplomacy is a board game that can serve as a sandbox, i.e., an abstract analog to real world domains, enabling us to explore this topic.

Our results on deviation from contracts consider multiple types of communicating agents, shown in Fig. 5 (in contrast to No-Press Diplomacy agents[47–49] such as the non-communicating baseline[39]). We call the agents of the section "Negotiating with binding agreements" Baseline Negotiators as they operate assuming agreements are binding. We consider Deviator Agents which overcome Baseline Negotiators by deviating from agreed contracts. The "Baseline negotiatiors are defeated by deviators who break contracts" section discusses Simple Deviators and Conditional Deviators, and shows they outperform the Baseline Negotiators. The "Mitigating the deviation problem: defensive agents" section considers Defensive Agents, such as the Binary Negotiators and Sanctioning Agents, which deter Deviators from breaking contracts while retaining the advantages stemming from communication. It also describes the Learned Deviator, a Deviator optimized against a population of Defensive Agents, showing that it learns to rarely break its contracts. The full algorithms appear in the "Methods" section). Our Defensive Agents reproduce human behaviors of ceasing to trust peers who break promises or sanctioning such deviations, inspired by work on the Evolution of Cooperation[57] (see comparison in Supplementary Note 7, as well as a discussion of differences between Diplomacy and repeated games). We examine the conditions for honest cooperation in the large-scale temporally extended setting of Diplomacy, complementing work on cooperation in repeated games, where multiple players repeatedly interact by playing an identical simple stage game with one another.

### Baseline negotiatiors are defeated by deviators who break contracts

We first show the advantage that Deviator Agents obtain by breaking contracts. Simple Deviators behave as if no contract was accepted even when agreements are reached. When a contract $D = (R_i, R_j)$ is reached, a Baseline Negotiator $p_i$ only selects actions in $R_i$, which do not violate the contract. In contrast, the Simple Deviator forgets the contract, and always samples actions from the unconstrained policy (possibly

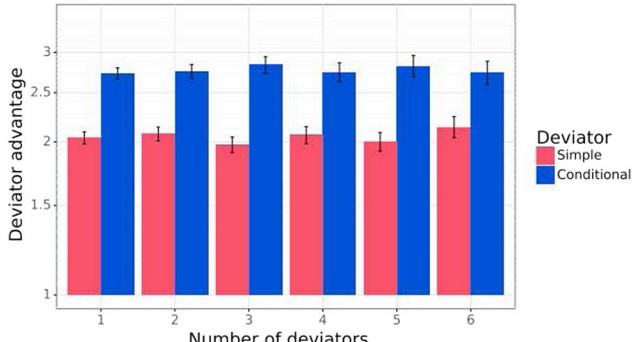
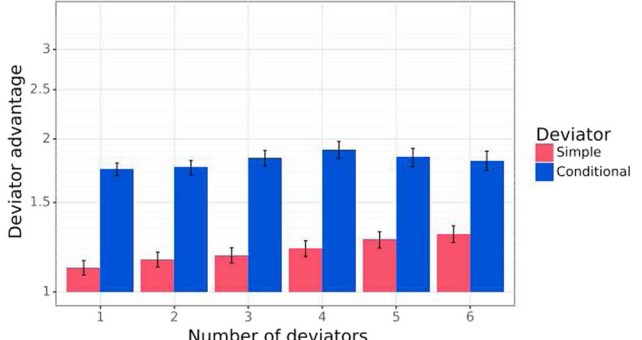

**Fig. 6 | Deviator agents playing versus Baseline Negotiator agents.** Left: Mutual Proposal protocol. Right: Propose-Choose protocol. The $x$ axis is the number of Deviator agents, and the $y$ axis is the winrate advantage of the deviator agent over the Baseline Negotiators, expressed as a ratio $\frac{W_D}{W_C}$, where $W_D$ is the average winrate of the Deviator agents and $W_C$ of the Baseline Negotiator agents, measured over 10,000 games.

selecting ones that violate the contract). Conditional Deviators are more sophisticated, and optimize their actions assuming that peers who accepted a contract would act according to it. Similarly to Simple Deviators, when no deal has been agreed, a Conditional Deviator $p_i$ selects actions from the unconstrained policy. However, when a deal $D = (R_i, R_j)$ is agreed on with another player $p_j$, the Conditional Deviator considers multiple actions it might take (these are sampled from its policy, and likely include candidate actions $a \notin R_i$ that violate the contract). For each such action, it performs multiple simulations of the next turn, by sampling actions for $p_j$ from the policy constrained by the contract, reflecting the assumption that $p_j$ would honor the agreement and thus refrain from taking an action $a \notin R_j$. The Conditional Deviator uses its average value in the sample to estimate the expected next turn utility for each candidate action, selecting the action that maximizes this expected value. The Deviator agents are fully described in the section "Deviator Agent algorithms".

We evaluate the relative performance of Baseline Negotiators and Deviators. Figure 6 shows the winrate ratio in games with $k$ Deviator agents playing against $7−k$ Baseline Negotiators, for $k \in \{1,2,...,6\}$. Figure 6 shows that even the Simple Deviator significantly outperforms the Baseline Negotiator, and that the Conditional Deviator overwhelmingly outperforms Baseline Negotiators (winning twice or three times more frequently).

### Mitigating the deviation problem: defensive agents
Communication allows Baseline Negotiators to outperform non-communicators, but the section "Baseline negotiatiors are defeated by deviators who break contracts" shows they are vulnerable to Deviators who gain the upper hand by breaking contracts. We overcome this problem using Defensive Agents that respond adversely to deviations. We consider Binary Negotiators that cease to communicate with agents who have deviated, and Sanctioning Agents who modify their goal to actively attempt to lower the deviator's value. We also consider Learned Deviators, which learn when to activate the deviator behavior mode (behaving identically to Baseline Negotiators until that moment); these agents learn to break agreements very rarely, making the vast majority of communication in the game truthful.

Defensive Agents deter deviations by negatively responding to them. Defensive agents initially act identically to the Baseline Negotiators, but when a defensive agent $p_i$ accepts a deal $D = (R_i, R_j)$ with a counterpart agent $p_j$, they examine the move that $p_j$ executes the next turn. If $p_j$ deviates from the agreement (i.e., takes an action $a_j \notin R_j$), they modify their behavior for the remainder of the game. We consider two responses to deviations: Binary Negotiators and Sanctioning Agents.

Binary Negotiators respond to a peer $p_j$ who deviates from an agreement by ignoring any communication from them until the end of the game. Following a deviation, a Binary Negotiator stops making any proposals to the deviator, and declines all proposals from the deviator (colloquially, they cease to trust the deviator).

A Sanctioning Agent $p_i$ responds to a deviation by a peer $p_j$ by selecting actions so as to lower the deviator $p_j$'s reward. A Baseline Negotiator $p_i$ evaluates actions sampled from the policy $\pi$ by performing simulations yielding possible future states $s'$, ranking actions by the expected future value $V_i(s')$. A Sanctioning Agent considers both its own value $V_i(s')$ and the deviator's value $V_j(s')$ to rank actions. It maximizes the metric $V_i(s') - \alpha V_j(s')$, which consists of both improving its own probability of winning $V_i(s')$ and lowering $p_j$'s probability of winning $V_j(s')$. The parameter $\alpha \geq 0$ controls the relative importance placed on the two goals, with $\alpha = 0$ reflecting no sanctioning (equivalent to Baseline Negotiators) and $\alpha = 1$ reflecting equal importance on lowering the Deviator's utility. As $\alpha \to \infty$, the Sanctioning Agent focuses solely on making the deviator lose the game, without regard to its own odds of winning the game (we use $\alpha = 1$, see Supplementary Note 10). Note that Sanctioning Agents themselves do not deviate from their contracts.

### Sanctioning dramatically reduces the advantage of deviation
We evaluate how a population of Defensive agents performs against Deviator agents. We consider games with $k$ Deviators playing against $7 − k$ Defensive agents, for $k \in \{1,2,...,6\}$, presented in Fig. 7. The figure shows that Binary Negotiators either outperform Deviators or least significantly reduce their advantage, and that Sanctioning Agents offer an even stronger defense, significantly outperforming deviators when the majority of the players are Sanctioning Agents. Defensive agents behave identically to Baseline Negotiators when playing with Baseline Negotiators, thus retaining the advantages of communication.

Hence, negatively responding to broken contracts allows agents to benefit from increased cooperation while resisting deviations. However, Deviators may adapt their behavior trying to render this defense less effective. For instance, a Simple or Conditional Deviator attempts to exploit others at every turn, selecting actions violating agreed contracts whenever they deem that doing so offers even a slight advantage, triggering the adverse response of Defensive agents early in the game. Hence, we consider Learned Deviators, whose parameters are optimized to best decide when to deviate in games against a population of Sanctioning agents.

A Learned Deviator $p_i$ considers two features of the current state $s$ and a contract $D = (R_i, R_j)$ agreed to with another player $p_j$: the approximate immediate deviation gain $\phi_i(s)$ reflecting the immediate improvement in value $p_i$ can gain by deviating from $D$, and the remaining opponent strength $\psi_i(s)$ reflecting the ability of the other agents to retaliate against $p_i$ should $p_i$ deviate from a contract. Given the state and the contracts agreed to by $p_i$, both of these features can be computed using the value and policy functions.

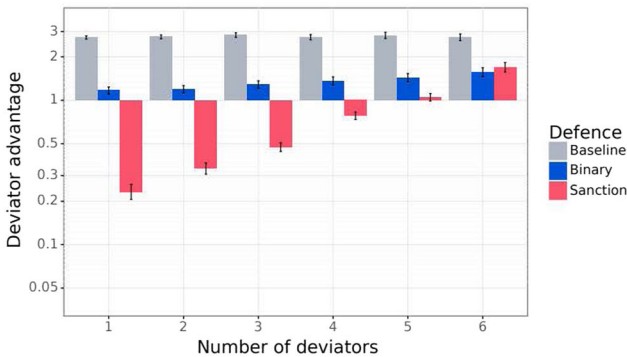 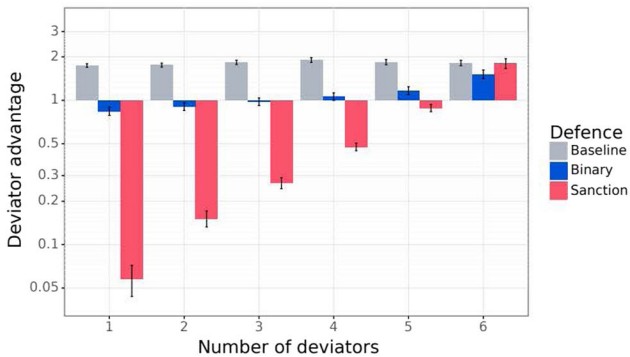

**Fig. 7 | Non-deviator agents (Baseline Negotiators, Binary Negotiators, and Sanctioning Agents) playing against Conditional Deviators.** Left: Mutual Proposal protocol. Right: Propose-Choose protocol. We consider multiple games, each with agents of exactly two types: either $k$ Deviators and $7 - k$ Baseline Negotiators, or $k$ Deviators and $7 - k$ Binary Negotiators, or $k$ Deviators and $7 - k$ Sanctioning Agents. The y-axis is the ratio $\frac{W_{dev}}{W_{def}}$ between the average winrate $W_{dev}$ of a Conditional Deviator and the average winrate $W_{def}$ of a Defensive agent (so values lower than 1 indicate a Defensive agent outperforms a Deviator agent). In both protocols,

a population of Binary Negotiators significantly lowers the advantage of Deviators, as compared to a population of Baseline Negotiators. While Deviators may still win more often than Binary Negotiators when there are few Binary Negotiators, the gap is much smaller than the one arising with Baseline Negotiators. Further, Sanctioning Agents offer a much stronger defense against Deviators. When there are more Sanctioning agents than Deviators, a Deviator wins the game much less frequently than a Sanctioning agent.

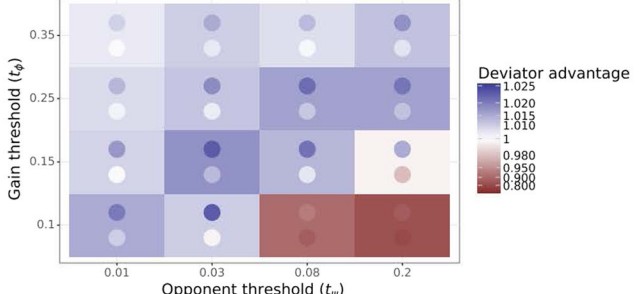 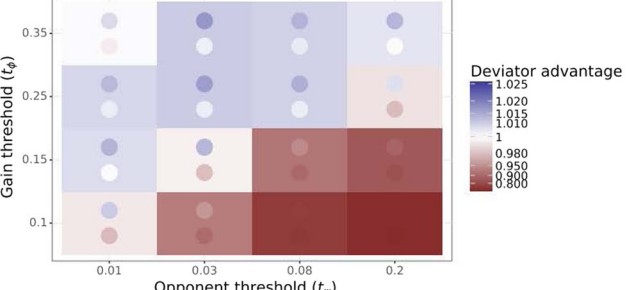

**Fig. 8 | Impact of the Learned Deviator parameters $(t_\phi, t_\psi)$ on the ratio $\frac{W_{dev}}{W_{san}}$ between the Deviator's winrate $W_{dev}$ and the Sanctioning Agent's winrate $W_{san}$, when playing against a population of 6 Sanctioning agents.** We call this the Deviator advantage. The cells are arranged along a horizontal $t_\psi$ axis and a vertical $t_\phi$ axis. The color of each cell indicates the Deviator advantage: blue indicates the advantage is >1 and, i.e., the Learned Deviator outperforms the Sanctioning Agents,

and red indicates the opposite. The colors of the top and bottom circles in each cell indicate the endpoints of a 95% confidence interval. These are evaluated with halved hyperparameters $M$ and $N$ relative to the earlier results. Left: Mutual Proposal protocol. Right: Propose-Choose protocol. Each cell is the result of over 40,000 games (with one Deviator using the given parameters, and 6 Sanctioning agents).

A Learned Deviator waits until a turn where the immediate gains from deviation are high enough and the ability of the other agent to retaliate is low enough, and only then deviates from an agreement. Such agents are parametrized by two thresholds $t_\phi, t_\psi$. They behave as a Baseline Negotiator, respecting all agreed contracts, until the first turn where the state $s$ is one such that $\phi(s) > t_\phi$ and $\psi(s) < t_\psi$. Then, they switch their behavior to that of the Conditional Deviator. Any two thresholds $t_\phi, t_\psi$ result in different Deviator behavior and winrate against a population of 6 Sanctioning agents. Full details regarding how the features $\phi(s), \psi(s)$ are computed and tuned to maximize the winrate against Sanctioning agents are given in the section "Learned Deviator". Figure 8 shows the winrate advantage of the Deviator for several choices of the two thresholds.

Even under the strongest parameter settings for the Learned Deviator, it only has a slight advantage, winning 1.7% more often than the Sanctioning agents with the Mutual Proposal protocol (1.0% for Propose-Choose). The average turn in which the Learned Deviator indeed deviates from an agreed contract is 82 (respectively, 93), while the average number of turns in games where deviations occurred is 101 (respectively, 110 for Propose-Choose). This indicates that the Learned Deviator adapts to break contracts quite late in the game. In the few games where the Learned Deviator does break its agreements, it often

wins: 53.1% of the time for Mutual Proposal and 52.6% for Propose-Choose (see Supplementary Note 9 for further behavioral analysis).

Overall across games, the Learned Deviator honors 99.8% (or 99.7% for Propose-Choose) of the contracts it had agreed to; by optimizing its behavior against a population of Sanctioning agents it adapts to honor the vast majority of its contracts, making the communication in these games almost entirely truthful.

## Discussion

We consider mechanisms enabling agents to negotiate alliances and joint plans. Similarly to recent work[25,27,31,58], we consider agents working in teams, each trying to counter the strategies of other teams. We take agents trained using reinforcement learning and augment them with a protocol for agreeing future moves. Our agents identify mutually beneficial deals by simulating future game states under possible contracts.

In terms of broader impact, Diplomacy[38] is a decades-long AI challenge, and we hope that this work will inspire future research on problems of cooperation. Diplomacy makes an exceptional testbed: it has simple rules but high emergent complexity, and an immense action space[40,43] which has recently been tackled using deep learning[39,47,48] (future work could of course uncover other successful approaches). Communication adds another important layer of

complexity. Game theory offers powerful tools for analyzing games such as Diplomacy and for constructing agents. In particular, solving for a subgame perfect equilibrium or applying repeated game analysis[7] could be useful for building capable agents. However, the large action space in Diplomacy results in an incredibly large game tree size, which impedes game theoretic analysis (see Supplementary Note 8). We thus combine reinforcement learning methods with game theoretic solutions (the Nash Bargaining Solution[52]), and address computational barriers using a Monte-Carlo simulation of potential future game trajectories.

Our results highlight the difficulty in establishing cooperation due to deviation from agreements. For humans, the potential to break promises or mislead others hinders cooperation, requiring people to decide whether they can trust others[50,51]. Similarly for AI agents in Diplomacy, when agreements are non-binding, an agent may promise to act one way during negotiation but act in a different way later. Our objective was to understand the conditions that foster truthful cooperation between artificial agents, and our results offer insight into some of the risks that arise with complex agents that can misrepresent intentions. We find that similarly to humans, cooperation is jeopardized by the ability of agents to break prior agreements. The "Baseline negotiatiors are defeated by deviators who break contracts" section shows that while Baseline Negotiators outperform silent agents due to increased cooperation, they are susceptible to peers who make false promises regarding future actions. We mitigate this risk by endowing agents with the inclination to retaliate against such deviations.

Strong Diplomacy AI gameplay may differ from human play[59] and different people may have different expectations regarding the behavior of artificial Diplomacy agents[60]. Recent methods recover human-like play through regularization[61]. We have not incorporated such regularization: after the policy network is learned from imitating human play, further training is based entirely on reinforcement learning.

Humans playing Diplomacy sometimes employ a strategy of keeping promises until a crucial deviation late in the game[62]. It's striking that our Learned Deviators follow a similar strategy: they adapt to typically follow through on their promises for long periods of time, breaking agreements only when the gains are high enough and when their opponents are left less capable of retaliating for the deviation. We strived to create a rich representation of the Diplomacy game, recreating its dilemmas relating to strategic communication. One possible cause for the strategic similarity is that Diplomacy was designed such that deviating from agreements can be beneficial when the opponent would find it difficult to respond. In other words, Diplomacy, even under the restricted communication protocols we considered, has a strategy space where such a behavior can be an effective way to win, so both humans and our Learned Deviator exhibit it.

The "Sanctioning dramatically reduces the advantage of deviation" section shows that when a population of agents are inclined to sanction broken agreements, they are resistant to deviators; when the proportion of retaliating agents is high enough, constant deviations from agreements are not beneficial as the long term negative consequences of the retaliation outweigh the immediate gains from the deviation. Even Learned Deviators, who optimize their choice of when to deviate against Defensive agents, rarely deviate from agreements. Hence, sanctioning peers who break contracts can foster more truthful communication among AI agents.

Learned Deviators do gain a slight advantage against Defensive agents, and when many agents deviate from agreements sanctioning deviations becomes more costly. Hence, Sanctioning agents may themselves start deviating from agreements, and if the sanctioning behavior proves to be costly, the Sanctioning agents may cease sanctioning deviators. In Supplementary Note 11 we probe the potential

learning dynamics of an agent population with regards to sanctioning and deviation: we consider Sanctioning agents who themselves deviate from agreements, and the incentive to cease sanctioning peers when this sanctioning behavior is costly. We find that a population of sanctioning agents is only at a near-equilibrium (in contrast to simple repeated games such as Iterated Prisoner's Dilemma which may admit a fully stable cooperative equilibrium[57]).

Such issues have been studied in the literature on the evolution of cooperation, and for simpler games researchers noted that additional mechanisms might be required to support sustained cooperation, such as having repeated interaction. Similarly for Diplomacy, to avoid a population of learning agents gradually becoming less cooperative one might need to employ additional mechanisms. Repeating the interaction and playing multiple games of Diplomacy against the same opponents could increase the incentives for cooperative behavior. Further, we focused on sanctioning, but humans rely on diverse solutions to deter deviating from agreements such as employing trust and reputation systems[63,64], or relying on a judicial system to provide remedies for the breach of contracts[65]. Future work could investigate how to implement such mechanisms for AI agents, similarly to earlier work on deterring deviations in dynamic coalition formation settings, for instance keeping track of peer behavior over multiple interactions or leveraging indirect reciprocity and tagging[66–68].

One limitation of our methods is that we took initial agents trained for No-Press Diplomacy, then augmented them with negotiation algorithms. This means that the action strategy, coming from the base agent, was designed separately from the communication and negotiation strategies. Future work might build stronger agents through better methods for co-evolving the action and communication strategies, allowing the agent's play to capture even more of the strategic richness that arises in human play. Our work has further limitations (discussed in depth in Supplementary Note 3), such as assuming a known deterministic model of the environment, holding the policy and value functions fixed following the initial reinforcement learning step, and using simple communication protocols rather than more elaborate ones or natural language. Further, many questions remain open for future research. Could one design more intricate protocols, considering more than two agents, or communicating knowledge or goals? How could one handle imperfect information? Finally, what other mechanisms could deter deviations from agreements?

## Methods
We present the algorithms for the agents of the sections "Negotiating with binding agreements" (Baseline Negotiator) and "Negotiating with non-binding agreements" (Simple, Conditional and Learned Deviators, Binary Negotiators and Sanctioning Agents).

### Simulation building blocks
Our methods use policies $\pi_i: S \times A_i \to [0,1]$ mapping any game state to a distribution over actions, with the probability of taking action $a \in A_i$ in state $s$ denoted as $\pi_i(s,a)$, and a state-value function $V: S \to \mathbb{R}^n$ which maps any state to the expected reward of all the agents (for Diplomacy this can be viewed as estimated win probabilities). We use $\pi$ to refer to policies $\pi_i$ for each player $p_i$. Various methods were proposed for constructing policy and value functions in Diplomacy[47–49]. Our experiments in "Results" use policy and value networks trained by first imitating human gameplay on a dataset of human Diplomacy games, then applying reinforcement learning to improve agent policies[39] (available at https://github.com/deepmind/diplomacy). Hence, the value and policy functions we use are based on both learning from human data and applying reinforcement learning, but our algorithms can work with any such functions. We denote sampling an action $a$ in state $s$ for agent $p_i$ from the policy $\pi_i$ as $a \sim \pi_i(s)$. Given a contract $D = (R_i, R_j)$ we can restrict the policies $\pi_i, \pi_j$ to respect the limitations expressed in the contract, yielding a policy $\pi_i^{R_i}$ (also written as $\pi_i^D$) for

$p_i$ where the probability of $p_i$ taking action $a$ in state $s$ is:

$$\pi_i^{R_i}(s,a) = \begin{cases} \frac{\pi_i(s,a)}{\sum_{a' \in R_i} \pi_i(a',s)}, & \text{for } a \in R_i \\ 0, & \text{for } a \notin R_i \end{cases} \qquad (1)$$

Note that given a neural network capturing a policy $\pi_i$ and a restricted action set $R_i$ (resulting from $p_i$ agreeing to a contract $D = (R_i, R_j)$), one can sample an action $a_i$ from $\pi_i^{R_i}(s)$ by masking the logits of all actions not in $R_i$ (i.e., setting the weight of all actions not in $R_i$ to zero before applying the final softmax layer). Similarly, we denote by $\pi_j^{R_j}$, or $\pi_j^D$, the policy for $p_j$ which respects the contract's restriction $R_j$.

## Estimating values using a sample of simulations

A key building block in our methods is to simulate what might occur in the next turn when players follow various policies, such as the unconstrained policy $\pi_i$ or a constrained policy $\pi_i^{R_i}$. Let $\mathbf{a}$ be a vector of agent actions $\mathbf{a} = (a_i)_{i=1}^n$, and let $T(s, \mathbf{a})$ be the transition function of the game, taking the current state $s$ and the action profile $\mathbf{a}$ and returning the resulting next game state. Finally, let $\mathbf{V}(s) = (V_i(s))_{i=1}^n$ be a state value function, denoting the expected reward (or probability of winning) of the players in a given state. We can combine these to obtain a joint action value function: $\mathbf{Q}(s, \mathbf{a}) = \mathbf{V}(T(s, \mathbf{a}))$.

Consider a player $p_i$ trying to select an action in a state $s$ and playing with agents $P_{-i} = \{p_1, \ldots, p_{i-1}, p_{i+1}, \ldots, p_n\}$ who follow the respective policies $\pi_{-i} = (\pi_1, \ldots, \pi_{i-1}, \pi_{i+1}, \ldots, \pi_n)$. Each policy in $\pi_{-i}$ may be for instance the unconstrained policy $\pi$ or alternatively $\pi_j$ may be a policy $\pi_j^{R_j}$ that is constrained by some contract $D = (R_j, R_k)$ that agent $p_j$ had agreed with some agent $p_k$. By $\mathbf{a} \sim \pi_{-i}$ we denote sampling actions for the players $P_{-i}$ from these respective policies $\pi_{-i}$. We denote the full action profile combining the action $a_i$ for $p_i$ and the actions $\mathbf{a}_{-i}$ for the players $P_{-i}$ as $(a_i, \mathbf{a}_{-i})$. By applying the joint action value function $\mathbf{Q}(s, (a_i, \mathbf{a}_{-i}))$ we arrive at an estimate of each player's value if $p_i$ takes the action $a_i$; we can consider this as a conditional action value function $\mathbf{Q}^{\mathbf{a}_{-i}}(s, a_i) = \mathbf{Q}(s, (a_i, \mathbf{a}_{-i}))$.

We can also extend this to two players: we denote the action profile combining $a_i$ for $p_i$ and $a_j$ for $p_j$ as $(a_i, a_j, \mathbf{a}_{-\{i,j\}})$, and the resulting pairwise conditional action value function as $\mathbf{Q}^{\mathbf{a}_{-\{i,j\}}}(s, (a_i, a_j)) = \mathbf{Q}(s, (a_i, a_j, \mathbf{a}_{-\{i,j\}})) = \mathbf{V}(T(s, (a_i, a_j, \mathbf{a}_{-\{i,j\}})))$.

Given a value function $V_i$, we can estimate $p_i$'s value for taking an action $a_i$ as the expected value $p_i$ in the next game state, with the expectation taken over the actions of the other agents sampled from their respective policies $\pi_{-i}$:

$$_i(s,a_i) = \mathbb{E}_{\mathbf{a}_{-i} \sim \pi_{-i}} V_i(T(s, (a_i, \mathbf{a}_{-i}))) = \mathbb{E}_{\mathbf{a}_{-i} \sim \pi_{-i}} Q_i^{\mathbf{a}_{-i}}(s, a_i) \qquad (2)$$

**Simulation action value estimation (SAVE).** Consider sampling a list of $M$ partial action profiles $\mathbf{a}_{-i}^1, \ldots, \mathbf{a}_{-i}^M \sim \pi_{-i}$. We denote this sample as $\mathbf{A}_{-i} = (\mathbf{a}_{-i}^1, \ldots, \mathbf{a}_{-i}^M)$ (where each individual element $\mathbf{a}_{-i}^m$ is a partial action *profile* that consists of actions for all players except $p_i$). We use the sample $\mathbf{A}_{-i}$ to get a Monte-Carlo estimate for agent $p_i$'s expected value when taking action $a_i$, by calculating $\hat{Q}_i^{\mathbf{A}_{-i}}(s, a_i) = \frac{1}{M} \sum_{m=1}^M Q_i^{\mathbf{a}_{-i}^m}(s, a_i)$. We refer to this as Simulation Action Value Estimation (SAVE). We may also estimate a different player $p_j$'s value $V_j(T(s, (a_i, \mathbf{a}_{-i})))$ when $p_i$ takes an action $a_i$ in a similar way. We use the notation $\hat{Q}_j^{\mathbf{A}_{-i}}(s, a_i)$. The same is true for the STAVE method which is introduced later.

**Simulation value estimation (SVE).** Consider a player $p_i$ considering a contract $D = (R_i, R_j)$. When $p_i, p_j$ select actions under this contract they use the respective policies $\pi_i^D = \pi_i^{R_i}$ and $\pi_j^D = \pi_j^{R_j}$. Assuming all other agents follow the unrestricted policy $\pi$, we consider the policy profile $\pi^D = (\pi_1^D, \ldots, \pi_n^D)$ where $\pi_i^D = \pi_i^{R_i}$ and $\pi_j^D = \pi_j^{R_j}$, and for any $k \notin \{i,j\}$ we

have $\pi_k^D = \pi_k$. Under these assumptions, we can estimate $p_i$'s value by taking the expected value in the next game state, with the expectation taken over all agents' actions when sampled from their respective policies in $\pi^D$:

$$_i^D(s) = \mathbb{E}_{\mathbf{a} \sim \pi^D} V_i(T(s, \mathbf{a})) = \mathbb{E}_{\mathbf{a} \sim \pi^D} Q_i(s, \mathbf{a}) \qquad (3)$$

We can obtain a Monte-Carlo estimate of the agents' values. We take a sample $\mathbf{A}$ of full action profiles $\mathbf{a}^1, \ldots, \mathbf{a}^M \sim \pi^D$ (where each $\mathbf{a}^m$ is a full action profile, consisting of actions for all the agents). Averaging over these, we obtain the value estimate $\hat{V}_i^{\mathbf{A}}(s) = \frac{1}{M} \sum_{m=1}^M Q_i(s, \mathbf{a}^m)$. We refer to this method as Simulation value estimation (SVE).

**Simulation two-action value estimation (STAVE).** We use a similar calculation for the Propose-Choose protocol, when considering the combined effect of an action $a_i$ for player $p_i$ and an action $a_j$ for player $p_j$, for example in order to evaluate a contract $D = (\{a_i\}, \{a_j\})$. In such situations, we may evaluate $p_i$'s value by taking their expected value in the next game state, over the actions of the other agents as sampled from their respective policies $\pi_{-\{i,j\}}$. This yields the following value:

$$_i(s, (a_i, a_j)) = \mathbb{E}_{\mathbf{a}_{-\{i,j\}} \sim \pi_{-\{i,j\}}} V_i(T(s, (a_i, a_j, \mathbf{a}_{-\{i,j\}}))) = \mathbb{E}_{\mathbf{a} \sim \pi^D} Q_i^{\mathbf{a}_{-\{i,j\}}}(s, (a_i, a_j)) \qquad (4)$$

This expectation is taken over many actions, so we approximate this using a Monte-Carlo simulation. We consider a sample $\mathbf{A}_{-\{i,j\}}$ consisting of actions for all players except $\{p_i, p_j\}$: $\mathbf{A}_{-\{i,j\}} = (\mathbf{a}_{-\{i,j\}}^1, \mathbf{a}_{-\{i,j\}}^2, \ldots, \mathbf{a}_{-\{i,j\}}^M)$ where each element $\mathbf{a}_{-\{i,j\}}^m$ contains the actions of all players except $p_i, p_j$. Given an action $a_i$ for agent $p_i$ and an action $a_j$ for $p_j$, each such element $\mathbf{a}_{-\{i,j\}}^m$ can be completed into a full action profile $(a_i, a_j, \mathbf{a}_{-\{i,j\}}^m)$, and used to evaluate the next state value.

Average over the sample we get: $\hat{Q}_i^{\mathbf{A}}(s, (a_i, a_j)) = \frac{1}{M} \sum_{m=1}^M Q_i^{\mathbf{a}_{-\{i,j\}}^m}(s, (a_i, a_j))$. We call this method Simulation two-action value estimation (STAVE).

## The sampled best responses procedure

A recent method for constructing No-Press Diplomacy agents starts with a policy which mimics human gameplay (imitation learning), then gradually improves this policy using a neural policy iteration process[39]. In the "Negotiators outperform non-communicating agents" section, we use agents produced under this approach as our baseline non-communicating agents. The neural policy iteration works by taking a current behavior policy network $\pi$, and applying an improvement operator aimed at generating actions that are better than those in this underlying policy. Many game trajectories are then generated using this improved policy. Given these games a new policy neural network $\pi'$ is distilled via supervised learning. The process is repeated, resulting in stronger and stronger policies.

We briefly describe the improvement operator used for No-Press Diplomacy[39], called Sampled Best Response (SBR for short), as we use it as a building block for our negotiating agents (see the recent paper[39] for further details about SBR and using it to derive Diplomacy RL agents). SBR selects the action of player $i$ in state $s$ as follows. SBR first samples many candidate actions $\{c^j\}_{j=1}^C$ for the target player $i$ from the policy $\pi^c$ (the current policy which SBR aims to improve upon). SBR then uses a policy $\pi^b$ to produce the actions of the other agents, either using the same network as $\pi^c$ or a previous generation policy.

To evaluate the quality of $c^j$, SBR simulates the possible behavior of other players by taking a sample $\mathbf{A}$ of partial action profiles $\mathbf{A} = \mathbf{a}_{-i}^1, \ldots, \mathbf{a}_{-i}^M \sim \pi_{-i}^b$, then evaluating the estimated future reward (akin to the SAVE procedure of the section "Simulation building blocks"):

$$\hat{Q}_i^{\mathbf{A}_{-i}}(s, c^j) = \frac{1}{M} \sum_{m=1}^M Q_i^{\mathbf{a}_{-i}^m}(s, c^j) = \frac{1}{M} \sum_{m=1}^M V_i(T(s, (c^j, \mathbf{a}_{-i}^m))) \qquad (5)$$

The action maximizing this estimated value under these simulations is returned by SBR. The SBR procedure[39] is given in Algorithm 1.

**Algorithm 1** Sampled Best Response.

```
1: function SBR(s: state, i: player, π^b_{-i}: peer policies, π^c_i: player policy, V_i: value function,
              M: number of peer action profiles, N: number of candidates)
2:    a^1_{-i}, ..., a^M_{-i} ∼ π^b_{-i}(s)                        ▷ Sample Action Profiles
3:    A_{-i} ← (a^1_{-i}, ..., a^M_{-i})
4:    c^1, ..., c^N ∼ π^c_i(s)                                   ▷ Sample Candidate Actions
5:    return argmax_{c∈{c^1,...,c^N}} Q̂^{A_{-i}}_i(s, c)
```

### Baseline Negotiator algorithms

In the following sections, we describe the algorithms for the Baseline Negotiator of the section "Negotiating with binding agreements", including RSS and MBDS. We first describe the action phase, as it is the same for both protocols. If no agreement has been reached during the negotiation phase in state $s$, the Baseline Negotiator $p_i$ is free to choose any legal action from its unconstrained policy $\pi^c_i$, and has no constraining information about other players' moves. It thus selects an action by applying the Sampled Best Response (SBR) method described in Algorithm 1, selecting the action $a_i = \text{SBR}(s, i, \pi^b_{-i}, \pi^c_i, V_i, M, N)$.

If a contract $D_{i,j} = (R_i, R_j)$ was agreed, it places a restriction $R_i$ that the Baseline Negotiator respects. If $R_i$ contains only a single action that $p_i$ may take under the contract, the Baseline Negotiator simply selects that action. If multiple actions are allowed under $R_i$, the Baseline Negotiator selects an action by applying SBR on a restricted policy $\pi^{R_i}$ and assuming that $p_j$ would also respect the agreement and select actions from the restricted policy $\pi^{R_j}$. If there are multiple agreed contracts $D_{i,j_1} = (R^{j_1}_i, R_{j_1}), ..., D_{i,j_k} = (R^{j_k}_i, R_{j_k})$ (as permitted by the Mutual Proposal protocol) then $R_i = R^{j_1}_i \cap \cdots \cap R^{j_k}_i$ is the intersection of the constraints. The negotiator then selects an action $a_i = \text{SBR}(s, i, (\pi^{R_{j_1}, b}_{j_1}, ..., \pi^{R_{j_k}, b}_{j_k}, \pi^b_{-\{i, j_1, ..., j_k\}}), \pi^{R_i, c}_i, V_i, M, N)$.

### Restriction simulation sampling (RSS)

Our negotiation algorithm for the Mutual Proposal protocol, called RSS, is based on applying SVE to contrast what might occur when a contract is agreed and when it is not agreed.

We consider the case where there is a single contract $D_{i,j} = (R_i, R_j)$ an agent may propose to another, such as a Peace contract in Diplomacy. Under the Mutual Proposal protocol an agent $p_i$ must decide whether to extend an offer to each of the other agents. When deciding whether agent $p_i$ would make a proposal $D_{i,j}$ to agent $p_j$, RSS uses SVE to estimate the expected value to $p_i$ in two cases: (1) assuming that $p_i$ does not reach an agreement with $p_j$, and (2) assuming $p_i$ and $p_j$ agree on the contract $D_{i,j} = (R_i, R_j)$. The full RSS method, given in Algorithm 2, compares agent utilities between these cases.

**Algorithm 2** Restriction Simulation Sampling.

```
1: function RSS(s: state, i: player, π: policies, V_i: value function, D_{i,1}, ..., D_{i,n}: contracts,
              M: number of action profiles)
2:    b^1, ..., b^M ∼ π
3:    d_1 = d_2 = ... = d_n = False
4:    for j ← 1 to n do
5:       (R_i, R_j) ← D_{i,j}
6:       if b^1_j, ..., b^M_j ∈ R_j then
7:          continue                                      ▷ Reject deal that may not constrain peer
8:       for m ← 1 to M do
9:          b'^m ← b^m
10:         if b^m_i ∉ R_i then                            ▷ Resample deal-violating actions
11:            b'^m_i ∼ π^{R_i}_i
12:         if b^m_j ∉ R_j then
13:            b'^m_j ∼ π^{R_j}_j
14:      B ← (b^1, ..., b^M)
15:      B' ← (b'^1, ..., b'^M)
16:      if V̂^{B'}_i(s) > V̂^B_i(s) then                    ▷ Compare SVEs
17:         d_j ← True
18:   return d_1, ..., d_n
```

For the first case, where no agreement is reached, both $p_i$ and $p_j$ are free to select any action, so we use the unrestricted policies $\pi_i$ and $\pi_j$ for $p_i$ and $p_j$. RSS makes the simplifying assumption that no other agent would reach any agreements (i.e no agent in the set $P\backslash\{p_i, p_j\}$ would reach an agreement with another agent), so we also use the unrestricted policies $\boldsymbol{\pi}$ for all the other agents. Hence, the case of not having an agreement is evaluated by applying SVE with the policy profile $\boldsymbol{\pi}$, so actions for all players $p_1, ..., p_n$ are sampled from their respective unconstrained policies $\pi_1, ..., \pi_n$.

In the second case, where an agreement $D_{i,j} = (R_i, R_j)$ is reached between $p_i, p_j$, both $p_i, p_j$ may only choose actions in the respective action sets $R_i, R_j$, so we use the restricted policies $\pi^{R_i}_i, \pi^{R_j}_j$ for $p_i$ and $p_j$ respectively. Similarly to the no agreement case, we make the simplifying assumption that no further contracts between other agents have been agreed, so we can reuse the actions for the other players $P\backslash\{i, j\}$ as used in the SVE procedure for the no-agreement.

To estimate values in the case of reaching an agreement we also require samples from $\pi^{R_i}_i$ and $\pi^{R_j}_j$ for $p_i$ and $p_j$; to get these, we reuse the constraint-satisfying actions from the no-agreement SVE, resampling only those actions which fall outside of $R_i$ and $R_j$. For instance, for a Peace contract, the resampling for $p_i, p_j$ is done by rejecting actions that violate the peace. We do so to reduce the variance in the estimate of the difference between values in the agreement and non-agreement case.

We denote by $\hat{V}^{\mathbf{B}}_i(s)$ the SVE value estimate for $p_i$ in the case of no agreement, and by $\hat{V}^{\mathbf{B}'}_i(s)$ the SVE estimate for $p_i$ in the case of an agreement $D = (R_i, R_j)$. If $\hat{V}^{\mathbf{B}'}_i(s) > \hat{V}^{\mathbf{B}}_i(s)$ then $p_i$ expects to achieve more utility when the contract is agreed, so they propose it to $p_j$, and otherwise they refrain from proposing it.

### Mutually beneficial deal sampling (MBDS)

Our negotiation algorithm for the Propose-Choose protocol is MBDS. It decides which contracts to propose during the Propose phase, and which contracts to choose during the Choose phase. It generates and selects contracts seeking mutually beneficial deals. In order to choose between possible contracts, we used the Nash Bargaining Solution (NBS) from game theory[52,53]. Given a space $S \subseteq \mathbb{R}^2$ of feasible agreements (with a point $s = (d_1, d_2)$ in $S$ yielding respective utilities $d_1, d_2$), and a disagreement outcome $d^0 = (d^0_1, d^0_2)$ (with respective utilities $d^0_1, d^0_2$), the NBS is the feasible agreement maximizing the product of utilities over the disagreement baseline: $\text{argmax}_{(d_1, d_2) \in S}(d_1 - d^0_1)^+ (d_2 - d^0_2)^+$.

NBS is the only bargaining solution concept that satisfies certain fair bargaining axioms, producing a negotiation outcome that's mutually advantageous relative to the no-agreement outcome[52].

NBS guarantees Pareto optimality, meaning negotiators never select a contract if they can find an alternative contract that guarantees both sides higher utility than the chosen agreement, independence of units of measure, meaning switching to an equivalent utility representation other than winrates does not result in a change in the chosen contracts, and symmetry, meaning that if the space of possible contracts is symmetric in the winrates it allows, then the chosen agreement would yield both sides the same winrate.

These properties of the NBS are important for our Diplomacy agents, as they guarantee that agreements are chosen so as to maximize the win probabilities of both sides and provide robustness to our choice of measuring utility in terms of winrate improvements. Further, these properties mean that negotiators attempt to choose balanced contracts that both sides may agree on, rather than choosing contracts that prefer one side over the other, which would make it more difficult to agree.

Approximating the Nash Bargaining Solution (NBS): When applying NBS to the Propose-Choose setting, one reasonable interpretation is to consider the disagreement utility as the win probability of the sides assuming no contract is agreed (with both sides selecting actions from their unconstrained policy), and to consider

the utility under a contract as the win probabilities of the sides assuming the contract is signed (with both sides selecting actions from the constrained policies under the contract).

However, Diplomacy has more than two agents and combinatorially large action spaces, and agent policies are stochastic. Diplomacy has an enormous action space of $10^{21}$ to $10^{64}$ legal actions per turn[39], and the space of possible contracts grows quadratically in the size of the legal actions set (any combination of a legal action for one side and a legal action for the other side makes a possible contract). Hence, it is intractable to exhaustively search through the space of contracts, or to exactly compute expected win probabilities under a contract. In addition, the interpretation of disagreement utility as assuming no contract is signed by either side is flawed: either side may instead sign a contract with a different peer. We overcome these difficulties using a sampling approach, and by simulating the negotiation process of other agents.

During the propose phase, MBDS generates multiple candidate contracts which agent $p_i$ could offer to $p_j$. The candidate contracts $\mathcal{D}_{i,j}$ are created by generating sets of candidate actions $C_i$ for $p_i$ and $C_j$ for $p_j$, and looking at the Cartesian product $C_i \times C_j = \{(c_i, c_j): c_i \in C_i, c_j \in C_j\}$ of these actions, i.e., all possible combinations of the candidate actions for the two sides. The candidate actions for the sides are generated by sampling many actions $c_i^1, ..., c_1^N$ and $c_j^1, ..., c_j^N$ from policies $\pi_i^c$ and $\pi_j^c$, and selecting the top $K$ ranked by a metric that combines both $p_i$'s and $p_j$'s expected utility under the actions (making the simplifying assumption that all remaining agents simply select an action using the unrestricted policy profile $\pi_{-\{i,j\}}^b$). This unrestricted policy is represented using a list of samples $\mathbf{b}^1, ..., \mathbf{b}^M \sim \pi^b$.

We consider these action profiles $\mathbf{B} = (\mathbf{b}^1, ..., \mathbf{b}^M)$, and an additional scaling factor $\frac{q_{i,j}^0}{q_{j,i}^0}$. Given these, the combined metric for an action $c_i$ is a weighted sum of $p_i$'s utility and $p_j$'s utility: $\hat{Q}_i^{\mathbf{B}-i}(s,c_i) + \beta \frac{q_{i,j}^0}{q_{j,i}^0} \hat{Q}_j^{\mathbf{B}-i}(s,c_i)$.

The parameter $0 < \beta < 1$ reflects the degree of emphasis on the utility of the partner to whom the contract is offered; low values of $\beta$ reflect emphasizing actions that lead to a high utility for the proposer, and high values of $\beta$ emphasize actions that lead to a high utility for the proposee.

The scaling factor $\frac{q_{i,j}^0}{q_{j,i}^0}$ adjusts the utilities to be on the same scale, facilitating deal-making between players with high versus low estimated values. It is computed by sampling a set $B$ of $M$ action profiles for all players from $\pi^b$, and using SAVE to select selfishly-best actions $c_i^*$ from among $c_i^1, ..., c_i^N$ and $c_j^*$ from among $c_j^1, ..., c_j^N$. $c_i^*$ and $c_j^*$ are then combined using STAVE to estimate $p_i$'s and $p_j$'s values in the case where no deal is made: $q_{i,j}^0 = \hat{Q}_i^{\mathbf{b}-\{i,j\}}(s,(c_i,c_j))$ and similarly for $q_{j,i}^0$.

When deciding about which of the contracts in $\mathcal{D}_{i,j}$ to propose to $p_j$, the proposer $p_i$ must consider and balance two factors relating to the resulting agent utilities (winrates): (1) the value that $p_i$ stands to gain should the contract be agreed on, and (2) the likelihood of $p_j$ agreeing to the contract, which is determined by the value $p_j$ stands to gain should the contract be agreed.

We combine these two factors by using the Nash Bargaining Solution[52], which reflects reasonable bargaining axioms[69]. We compute the Nash Bargaining Score in Algorithm 3, which relies on $q_i, q_j$ which are used as estimates of the no-deal baseline values that the gains are relative to. The values $q_i, q_j$ indicate what values $p_i$ and $p_j$ may expect to obtain in the absence of a deal between them.

**Algorithm 3** Nash Bargaining Score.

1: **function** NASHSCORE($s$: state, $i,j$: players, $c_i \in A_i, c_j \in A_j$: contract actions, $q_i, q_j$: baseline values, $\mathbf{B}_{-\{i,j\}}$: partial action profiles)
2:   **return** $\max\left(0, \hat{Q}_i^{\mathbf{B}-\{i,j\}}(s,(c_i,c_j)) - q_i\right) \cdot \max\left(0, \hat{Q}_j^{\mathbf{B}-\{i,j\}}(s,(c_i,c_j)) - q_j\right)$

When there is no agreement, both $p_i$ and $p_j$ are free to choose any legal action. Hence, we assume they would choose the best

actions $c_i^*$ and $c_j^*$ that they can from among samples from the policy $\pi^c$. As in SBR, we take the simplifying assumption that that the remaining agents $P \backslash \{p_i, p_j\}$ would not form agreements amongst themselves, and hence would simply choose actions from their policy $\pi^b$. Seeking to estimate the next state values for this case, we use the STAVE algorithm described earlier, defining $q_{i,j}^0 = \hat{Q}_i^{\mathbf{B}-\{i,j\}}(s,(c_i,c_j))$ and $q_{j,i}^0 = \hat{Q}_j^{\mathbf{B}-\{i,j\}}(s,(c_i,c_j))$. We refer to these values as the initial no-deal value estimates.

The initial no-deal estimates may not fully represent the values $p_i$ and $p_j$ can reach without a deal between them, as either side may be able to do better by making a deal with another player. To estimate this we use an iterative approach akin to other iterative negotiation methods[70] and described in Algorithm 4.

**Algorithm 4** BATNA Update by Internal Dynamic Bargaining Simulation.

1: **function** MBDS-BARGAIN($s$: state, $\mathbf{B}$: action profiles, $(\mathcal{D}_{i,j})_{i,j=1}^n$: candidate deals, $(q_{i,j}^0)_{i,j=1}^n$: no-deal BATNA value estimates, $(q_{i,j})_{i,j=1}^n$: BATNA value estimates, $\kappa$: damping factor)
2:   **for** $i \leftarrow 1$ to $n$ **do**
3:     **for** $j \leftarrow 1$ to $n$, $j \neq i$ **do**                          ▷ Find Nash-Bargaining Deal with $j$
4:       $(\{c_i\}, \{c_j\}) \leftarrow \text{argmax}_{(\{c_i\},\{c_j\}) \in \mathcal{D}_{i,j}} \text{NASHSCORE}(s,i,j,c_i,c_j,q_{i,j},q_{j,i},\mathbf{B}_{-\{i,j\}})$
5:       $r_j \leftarrow \hat{Q}_i^{\mathbf{B}-\{i,j\}}(s,(c_i,c_j))$
6:       **if** $\text{NASHSCORE}(s,i,j,c_i,c_j,q_{i,j},q_{j,i},\mathbf{B}_{-\{i,j\}}) = 0$ **then** $r_j \leftarrow -\infty$
7:     **for** $j \leftarrow 1$ to $n$, $j \neq i$ **do**
8:       $q_{i,j}' \leftarrow \max(q_{i,j}^0, \max_{k \in \{1,...,n\} \backslash \{i,j\}} r_k)$         ▷ Find BATNA
9:       $q_{i,j}' \leftarrow \kappa q_{i,j} + (1-\kappa) q_{i,j}'$                             ▷ Apply Damping
10:  **return** $(q_{i,j}')_{i,j=1}^n$

Algorithm 4 simulates an iterative process of updating beliefs regarding the contracts that might be agreed between all agents, and the resulting estimates of the values all agents might obtain. In each iteration, we examine each pair of players, and assume the deal they would select is the one with the highest Nash Bargaining Score over the current baseline estimates. Further, the update assumes that when each player $p_i$ considers their action with regard to each partner $p_j$, it contrasts its no-deal baseline with $p_j$ with the best alternative deal from among the remaining players $P \backslash \{p_i, p_j\}$. We apply a damping factor $\kappa$ to avoid oscillations in partner choice. Using the updated baselines, MBDS proposes to each partner the deal with the highest score. The full procedure for the Proposal phase is given in Algorithm 5.

**Algorithm 5** Mutually Beneficial Deal Sampling Proposal.

1: **function** MBDS-PROPOSE($s$: state, $i$: player, $\pi^c$: candidate policies, $\pi^b$: peer policies, $\mathbf{V}$: value functions, $M$: number of action profiles, $N$: number of sampled candidates, $K$: number of deal candidates, $\beta$: partner value weight, $R$: number of bargain rounds, $\kappa$: bargain damping factor)
2:   $\mathbf{b}^1, ..., \mathbf{b}^M \sim \pi^b$
3:   $\mathbf{B} \leftarrow (\mathbf{b}^1, ..., \mathbf{b}^M)$
4:   **for** $j \leftarrow 1$ to $n$ **do**
5:     $c_j^1, ..., c_j^N \sim \pi_j^c$                                           ▷ Sample Candidates
6:     $c_j^* \leftarrow \text{argmax}_{c \in \{c_j^1,...,c_j^N\}} Q_j^{\mathbf{B}-j}(s,c)$          ▷ Best Responses to Base Profiles
7:   **for** $j \leftarrow 1$ to $n$, $k \leftarrow 1$ to $n$ **do**
8:     $q_{j,k}^0 \leftarrow \hat{Q}_j^{\mathbf{B}-\{j,k\}}(s,(c_j^*,c_k^*))$              ▷ Initial BATNA Value Estimates
9:   **for** $j \leftarrow 1$ to $n$, $k \leftarrow 1$ to $n$ **do**
10:    $C_{j,k} \leftarrow \left\{\text{top } K \text{ actions } c \in \{c_j^1,...,c_j^N\} \text{ ranked by } \hat{Q}_j^{\mathbf{B}-j}(s,c) + \beta \frac{q_{j,k}^0}{q_{k,j}^0} \hat{Q}_k^{\mathbf{B}-j}(s,c)\right\}$
11:  **for** $j \leftarrow 1$ to $n$, $k \leftarrow 1$ to $n$, $j \neq k$ **do**
12:    $\mathcal{D}_{j,k} = C_{j,k} \times C_{k,j}$                                    ▷ Candidate Deals
13:  **for** $r \leftarrow 1$ to $R$ **do**                                    ▷ Update BATNA Estimates
14:    $(q_{j,k}^r)_{j,k=1}^n \leftarrow \text{MBDS-BARGAIN}(s,(\mathbf{b}^m)_{m=1}^M,(\mathcal{D}_{j,k})_{j,k=1}^n,(q_{j,k}^0)_{j,k=1}^n,(q_{j,k}^{r-1})_{j,k=1}^n,\kappa)$
15:  **for** $j \leftarrow 1$ to $n$ **do**
16:    $D_j \leftarrow \text{argmax}_{(\{c_i\},\{c_j\}) \in \mathcal{D}_{i,j}} \text{NASHSCORE}(s,i,j,c_i,c_j,q_{i,j}^R,q_{j,i}^R,\mathbf{B}_{-\{i,j\}})$
17:    $(c_i,c_j) \leftarrow D_j$
18:    **if** $\text{NASHSCORE}(s,i,j,c_i,c_j,q_{i,j}^R,q_{j,i}^R,\mathbf{B}_{-\{i,j\}}) = 0$ **then** $D_j \leftarrow \emptyset$
19:  **return** $D_1, ..., D_{i-1}, D_{i+1}, ..., D_n$

MBDS applies the same ranking during the Choose phase. In this phase, each agent must choose from amongst the contracts that are On The Table for them, selecting a single partner to agree a contract with. MDBS for a target agent $p_i$ examines the set of contracts involving $p_i$, and computes the Nash Bargaining score for each one using the baseline values from the Proposal phase. If some Nash Bargaining scores are positive, it considers the partners with whom such deals are possible, selecting the partner with whom the highest-score deal is most favorable to itself. If not, it simply selects the deal most favorable

to itself from among those that are estimated to be preferable to the no-deal baseline with their respective counterpart.

Lastly, for the chosen deal, MBDS evaluates and indicates whether both deals with the chosen counterpart would be acceptable, by comparing their estimated values with the no-deal BATNA. The full MBDS method for the Choose phase is given in algorithm 6.

**Algorithm 6** Mutually Beneficial Deal Sampling Choice.

```
 1: function MBDS-CHOOSE(s: state, i: player, B: action profiles, V: value functions,
                        (q_{i,j}^0, q_{j,i}^0)_{j=1}^n: no-deal BATNA value estimates,
                        (q_{i,j}^R, q_{j,i}^R)_{j=1}^n: BATNA value estimates, (D_{i,j}, D_{j,i})_{j=1}^n: proposed deals)
 2:     D* ← ∅
 3:     v* ← −∞
 4:     for j ← 1 to n, j ≠ i do                                    ▷ Identify Best Fair Deal
 5:         bothAcceptable_j ← (q_{i,j}^0 < min_{({c_i},{c_j})∈{D_{i,j},D_{j,i}}\{∅}} Q̂_i^{B−{i,j}}(s,(c_i,c_j)))
 6:         ({c_i},{c_j}) ← argmax_{({c_i},{c_j})∈{D_{i,j},D_{j,i}}\{∅}} NASHSCORE(s,i,j,c_i,c_j,q_{i,j}^R,q_{j,i}^R,B_{−{i,j}})
 7:         v ← Q̂_i^{B−{i,j}}(s,(c_i,c_j))
 8:         if v > max(v*,q_{i,j}^0) and NASHSCORE(s,i,j,c_i,c_j,q_{i,j}^R,q_{j,i}^R,B_{−{i,j}}) > 0 then
 9:             D* ← ({c_i},{c_j})
10:             v* ← v
11:             j* ← j
12:     if D* ≠ ∅ then
13:         return D*, bothAcceptable_{j*}
14:     for j ← 1 to n, j ≠ i do                                  ▷ Identify Best Favourable Deal
15:         ({c_i},{c_j}) ← argmax_{({c_i},{c_j})∈{D_{i,j},D_{j,i}}\{∅}} Q̂_i^{B−{i,j}}(s,(c_i,c_j))
16:         v ← Q̂_i^{B−{i,j}}(s,(c_i,c_j))
17:         if v > max(v*,q_{i,j}^0) then
18:             D* ← ({c_i},{c_j})
19:             v* ← v
20:             j* ← j
21:     return D*, bothAcceptable_{j*}
```

## Deviator Agent algorithms

Deviators are agents who may deviate from deals they have agreed to. In the following sections, we fully describe the Simple Deviators and Conditional Deviators of the section "Negotiating with non-binding agreements". Simple Deviators enter into deals in a way that is identical to the Baseline Negotiators, but even when a deal is agreed on they select their action directly from the unconstrained SBR policy, as if no deal was agreed. Conditional Deviators also enter into deals identically to Baseline Negotiators, but when selecting their action in turns when a deal has been reached, they attempt to maximize their reward under the assumption that the other side would honor the deal.

## Simple Deviators

During the negotiation phase, Simple Deviators behave identically to the Baseline Negotiator. Simple Deviators only behave differently to Baseline Negotiators during the action phase.

During the action phase in state $s$, if no deal has been reached for a Baseline Negotiator $p_i$, this Baseline Negotiator selects an action $a_i = \text{SBR}(s,i,\pi_{-i}^b,\pi_i^c,V_i,M,N)$ using the unconstrained policy $\pi_i^c$. However, if the Baseline Negotiator $p_i$ agreed to deals $D_{i,j} = (R_i^j,R_j)$ with players $j_1,...,j_k$ then the Baseline Negotiator $p_i$ selects an action $a_i = \text{SBR}(s,i,(\pi_{j_1}^{R_{j_1},b},...,\pi_{j_k}^{R_{j_k},b},\pi_{-\{i,j_1,...,j_k\}}^b),\pi_i^{R_i^{j_1}∩···∩R_i^{j_k},c},V_i,M,N)$ using the constrained policies $\pi_i^{R_i^{j_1}∩···∩R_i^{j_k},c}$ and $\pi_j^{R_j^{j_1},b}$.

In contrast to a Baseline Negotiator, a Simple Deviator $p_i$ always selects an action from the unconstrained policy $\pi_i^c$, regardless of whether a deal has been agreed or not (i.e., it always selects an action $a_i = \text{SBR}(s,i,\pi_{-i}^b,\pi_i^c,V_i,M,N)$). Colloquially speaking, we say that Simple Deviators forget that they have signed deals with others—even when a Simple Deviator $p_i$ has agreed on a deal $D = (R_i,R_j)$ with another player $p_j$, during the action phase it still chooses an action which does not necessarily respect the constraint $R_i$, and doesn't assume that $p_j$ will respect the constraint $R_j$.

## Conditional Deviators

Similarly to the Simple Deviators, during the negotiation phase the behavior of a Conditional Deviator is identical to that of a Baseline Negotiator. Conditional Deviators only behave differently to Baseline Negotiators during the action phase.

When no deal has been agreed, a Conditional Deviator $p_i$ selects an action in the same way as the Baseline Negotiator, i.e., using the SBR logic: sampling a set $\mathbf{B}_{-i}$ of $M$ partial action profiles, sampling $N$ candidate actions from the unrestricted policy $\pi_i^c$, and choosing the one yielding the maximal value on the sample, i.e., $\text{argmax}_{c∈\{c^1,...,c^N\}} \hat{Q}_i^{\mathbf{B}_{-i}}(s,c)$. However, when a deal $D = (R_i,R_j)$ has been agreed between a Conditional Deviator $p_i$ and some other player $p_j$, the Conditional Deviator attempts to maximize its gain under the assumption that the other side would respect their end of the deal; again we use a sample $\mathbf{B}_{-i}$ of $M$ partial action profiles, but in this sample we select the action for any deal partner $p_j$ from the restricted policy $\pi_j^{R_j,b}$, reflecting the assumption that the peer $p_j$ would behave according to the restriction of the contract. We consider the top candidate action sampled from the unconstrained policy $c^* = \text{argmax}_{c∈\{c^1,...,c^N\}} \hat{Q}_i^{\mathbf{B}_{-i}}(s,c)$. We then consider actions which do not violate any contract $p_i$ has agreed to, by taking a sample of candidate actions $c'^1,...,c'^N$ which are sampled from the policy $\pi_i^{R_i^1∩···∩R_i^n,c}$ that respects all the contracts that $p_i$ had agreed to. The top ranked action that does not violate $p_i$'s agreements is $c'^* = \text{argmax}_{c'∈\{c'^1,...,c'^N\}} \hat{Q}_i^{\mathbf{B}_{-i}}(s,c')$. If the estimated value of $c^*$ exceeds that of $c'^*$, the Conditional Deviator expects to gain by breaking its agreements and it chooses the contract-violating action $c^*$; otherwise, when a deviation is not deemed profitable, it selects the top contract-respecting action $c'^*$. This process is given in Algorithm 7.

**Algorithm 7** Conditional Deviator Action Selection for $p_i$.

```
 1: function CD-AS(s: state, i: player, π_i^c: candidate policy, V_i: value function, π^b: peer policies,
                  M: number of action profiles, N: number of sampled candidates,
                  D_{i,1},...,D_{i,n}: agreed contracts)
 2:     for j ← 1 to n do                                         ▷ Sample peer actions
 3:         if D_{i,j} ≠ ∅ then
 4:             (R_i^j,R_j) ← D_{i,j}
 5:         else
 6:             (R_i^j,R_j) ← (A_i,A_j)                           ▷ Unrestricted action space
 7:         b_j^1,...,b_j^M ∼ π_j^{R_j,b}
 8:     B_{−i} ← ((b_1^1,...,b_{i−1}^1,b_{i+1}^1,...,b_n^1),...,(b_1^M,...,b_{i−1}^M,b_{i+1}^M,...,b_n^M))
 9:     c^1,...,c^N ∼ π_i^c                                       ▷ Sample potential actions
10:     c^* ← argmax_{c∈{c^1,...,c^N}} Q̂_i^{B−i}(s,c)
11:     if D_{i,1} = D_{i,2} = ··· = D_{i,n} = ∅ then
12:         return c^*
13:     c'^1,...,c'^N ∼ π_i^{R_i^1∩···∩R_i^n,c}
14:     c'^* ← argmax_{c'∈{c'^1,...,c'^N}} Q̂_i^{B−i}(s,c')
15:     φ ← Q̂_i^{B−i}(s,c^*) − Q̂_i^{B−i}(s,c'^*)
16:     if φ > 0 then
17:         return c^*
18:     else
19:         return c'^*
```

## Learned Deviator

We describe our approach for training a deviator agent that learns the optimal state in which to deviate from a contract when playing a population of other agents (in our empirical analysis these peers are the Sanctioning agents of the section "Mitigating the deviation problem: defensive agents"). Consider that the defensive agents of the section "Mitigating the deviation problem: defensive agents" modify their behavior following any deviation from a previous agreement, and starting from that point they persist with their negative behavior towards the deviator for the remainder of the game; if $p_i$ has deviated from a deal with $p_j$ at a certain point in a game, $p_j$ will never again reach an agreement with $p_i$ in that game. Thus, a deviator agent $p_i$ playing against a defensive agent $p_j$ only has a single opportunity to deviate from an agreement with them in a game. Hence, it makes sense to pose the issue of optimizing the behavior of a deviator $p_i$ as identifying the point in the game where it is best to deviate from an agreement with some defensive agent $p_j$. We simplify the problem further by having the Learned Deviators consider only two features of the current state $s$

and contract $D = (R_i, R_j)$ agreed with a peer $p_j$. The first is the approximate immediate deviation gain $\phi = \hat{Q}_i^{\mathbf{B}_{-i}}(s, c^*) - \hat{Q}_i^{\mathbf{B}_{-i}}(s, c^{*'})$, as defined on line 15 of Algorithm 7, reflecting the immediate improvement in value $p_i$ can gain by deviating from $D$. The second is the strength of the other agents that $p_i$ may turn against itself should $p_i$ deviate from a contract. In Diplomacy, the win probability of a player $p_i$ in a state $s$ is closely correlated with the number of units (or supply centers) that this player has, and how well positioned they are to attack other players. These characteristics of the state are also closely correlated with the ability of that player to retaliate against an attack from another player. In other words, when a player $p_i$ considers deviating from an agreement with player $p_j$, they should be more worried regarding player $p_j$'s retaliation when $p_j$ has a high win probability (equivalently a high value) in the resulting state. Thus, we derive a measure $\psi$ from two considerations. One is that if none of the partners with whom a contract has been made is strong, $p_i$ may more safely deviate. The other is that if one of the partners is very dominant among the non-$p_i$ players, then that partner may be mostly incentivized to reduce $p_i$'s value to begin with, and so $p_i$ doesn't have as much to lose by triggering its retaliation. Thus, with $c^*$, $s$, $\mathbf{B}$, and $D_{i,j}$ as defined in Algorithm 7, we define $\psi = \min(\hat{Q}_{j^*}^{\mathbf{B}_{-i}}(s, c^*), 1 - \hat{Q}_i^{\mathbf{B}_{-i}}(s, c^*) - \hat{Q}_{j^*}^{\mathbf{B}_{-i}}(s, c^*))$, where $j^* = \text{argmax}_{j:D_{i,j} \neq \emptyset} \hat{Q}_j^{\mathbf{B}_{-i}}(s, c^*)$.

Finally, we define a Deviator agent $p_i$ parameterized by two thresholds $t_\phi, t_\psi$. It follows the Baseline Negotiator behavior (as though $\phi < 0$ in Algorithm 7) until the first turn where $\phi > t_\phi$ and $\psi < t_\psi$. Then, the agent switches its behavior to that of the Conditional Deviator. In other words, this Deviator waits until a turn where the immediate gains from deviation are high enough and the ability of the other agent to retaliate is low enough, and only then deviates from an agreement. In our experiments the thresholds $t_\phi, t_\psi$ are tuned by applying a simple grid search over the space of parameters (see Fig. 8), though many other approaches could be used (such as Gradient Ascent or Simulated Annealing).

**Defensive agent algorithms**
Defensive agents are designed to deter others from deviating from agreements, while still retaining the advantage negotiation skills afford over non-communicating agents; they shun Deviators, or work as a group to repel them by negatively responding to deviations so as to reduce the gains from such deviations. Defensive agents act identically to the Baseline Negotiator agents described in detail in the "Baseline Negotiator algorithms" section, but change their behavior when another agent deviates from an agreement made with them. When a defensive agent $p_i$ accepts a deal $D = (R_i, R_j)$ with a counterpart agent $p_j$, it examines the behavior of the counterpart $p_j$ in the following action phase. If $p_j$ deviates from the agreement by taking a disallowed action $a_j \notin R_j$, it changes its behavior towards this deviator for the remainder of the game. We now fully describe the Binary Negotiators and Sanctioning agents discussed in the section "Negotiating with non-binding agreements".

Binary Negotiators respond to a deviation of $p_j$ by ceasing all communication with $p_j$ for the remainder of the game. In the Mutual Proposal protocol this means never proposing any contract to $p_j$ and hence never agreeing on any contract with $p_j$ for the remainder of the game, and in the Propose-Choose protocol this means never choosing any contract proposed by or to $p_j$, hence never agreeing a contract with $p_j$ for the remainder of the game. This is done via an additional check on lines 8 and 17 of Algorithm 6.

A Sanctioning agent $p_i$ takes a more active role when responding to a deviation by player $p_j$. Following such a deviation, the Sanctioning agent $p_i$ modifies its behavioral goal so as to attempt to lower the deviator $p_j$'s reward in the game.

The standard SBR procedure, Algorithm 1 described in the section "The sampled best responses procedure", estimates the expected value for player $p_i$ in the next state when taking action $a$ as $\hat{Q}_i^{\mathbf{B}_{-i}}(s, a)$, and selects the actions that maximize this expected reward. Following a deviation by player $p_j$ against player $p_i$, we consider the modified score $\hat{Q}_{i-\alpha j}^{\mathbf{B}_{-i}}(s, a) = \hat{Q}_i^{\mathbf{B}_{-i}}(s, a) - \alpha \hat{Q}_j^{\mathbf{B}_{-i}}(s, a)$. This perturbed score represents a combination of the target player $p_i$'s utility and the negative of player $p_j$'s utility, where the parameter $\alpha$ controls the degree of importance placed on lowering player $p_j$'s utility.

A Sanctioning Agent $p_i$ selects actions identically to the Baseline Negotiator of the section "Baseline Negotiator algorithms" until a deviation occurs by a peer $p_j$. Following this point and until the end of the game, the Sanctioning Agent turns to selecting the actions maximizing the perturbed score $\hat{Q}_{i-\alpha j}^{\mathbf{B}_{-i}}$ defined above (in the negotiator algorithms, this affects the evaluation in RSS and on lines 5, 7, and 16 of the Choose phase of MBDS, but not in the Proposal phase). In the case where multiple players $p_{j^1}, \ldots, p_{j^k}$ have deviated from contracts with $p_i$, the negative utilities are combined to form $\hat{Q}_{i-\alpha j^1 - \cdots - \alpha j^k}^{\mathbf{B}_{-i}}$, defined analogously. Thus, following a deviation by player $p_j$, the Sanctioning agent $p_i$ optimizes for a perturbed score that represents a combination of maximizing the win probability of $p_i$ and minimizing the win probability of the deviator $p_j$.

## Data availability
The raw data for producing the figures is provided as Source data. Source data are provided with this paper.

## Code availability
The experiments are based on the algorithms contained in the "Methods" section, which are presented in more detail using Python-style code snippets in Supplementary Note 12. These depend on the base No-Press Diplomacy policy and value functions[39], which are available at https://github.com/deepmind/diplomacy.

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

## Acknowledgements

We would like to thank Will Hawkins, Aliya Ahmad, Dawn Bloxwich, Lila Ibrahim, Julia Pawar, Sukhdeep Singh, Tom Anthony, Kate Larson, Julien Perolat, Marc Lanctot, Edward Hughes, Richard Ives, Karl Tuyls, Satinder Singh and Koray Kavukcuoglu for their support and advice throughout the work.

## Author contributions

J.K. and Y.B. conceived the project, with support from the other authors. J.K. and T.E. developed the codebase, with support from the other authors. J.K., I.G., A.T., K.M., M.M., and T.G. decided on the the empirical analysis of the agent behavior and conducted it with support from the other authors. All authors contributed technical advice and ideas. Y.B. and J.K. wrote the initial draft of the paper, and all authors edited and refined the paper.

## Competing interests

The authors declare no competing interests.
