## [Peer Review File · Nature Communications]

REVIEWER COMMENTS

Reviewer #1 (Remarks to the Author):

This paper is amazing. It takes Diplomacy as sample problem for teaching agents how to negotiate non-binding agreements (as common in Diplomacy), and when to keep them or not keep them according to the overall chances of winning the game. In many ways, things nicely fall into place here, and it seems in the rearview so logical and clear. Which is, I think, good, because it shows that something has been achieved. In many ways, the sequence of agents that is presented reminds me of the hide-and-seek paper. And there is certainly room for more in this direction, but not all in a single paper. This is definitely a big step (whatever limited in means of communication) into the direction of seeing flexible communication mechanisms evolved in agents. Actually, the next steps then may be to make the "characters" even more flexible (more types allowed in Figure 5), and eventually mix them in the sense that, as humans do, the agents can adapt their strategies to the environment. Which is of course more difficult to sort out and investigate than the defined single types.

This direction (or whatever the authors think should be the next steps) may be made a bit more clearer in the intro already.

The most stunning surprise I had when reading the paper was that the strategies of the successful agents in the end very closely resemble successful human strategies. That is also an aspect that may be referred to more in the paper. Apart from the fact that it is quite rare that single players win the game when humans play, it is well known that a "buildup trust, deviate once, when it really leads to winning the game quickly" strategy is quite usual for human players. Which is very similar to the latest agents you present. It is very interesting to see that albeit partly different conditions (e.g. the amount of negotiations humans can actually do in the usually fixed time frame for negotiations is fairly limited), still very similar successful strategies "evolve". Did you drive it that way? Or did it just come out of how the first agents were "complexified"?

My expertise is more in the game AI direction but also in Diplomacy as a human player, not so much in general computational negotiation mechanisms, it may thus be that there is more from that field that shall be cited. But otherwise, references look good to me. There are some more papers on computational Diplomacy, I sense they are hard to find because "diplomacy" is not a good search term. Two that come to mind you may find interesting (up to you) are from this team (I have been evolved in these, so it is totally up to your choice if you want to take them into account):

Markus Kemmerling, Niels Ackermann, Nicola Beume, Mike Preuss, Sebastian Uellenbeck, Wolfgang Walz:

Is human-like and well playing contradictory for Diplomacy bots? CIG 2009: 209-216

The authors find that if bots play too good (restricted press), they may be seen as too far from human play.

Markus Kemmerling, Niels Ackermann, Mike Preuss:

Making Diplomacy Bots Individual. Believable Bots 2012: 265-288

This is somewhat similar, but finds that the expectations of humans how to actually identify bots compared to players are very different. Maybe not that surprising, but effectively that means that bots who may be seen as bots by one human are not recognized by another player because she/he has different expectations.

Overall, observations concerning the "strategy space" may be interesting. As already stated above, it looks as through subsequent experimentation with different strategies, a human-like playing style has evolved that seems superior at least on average (learned deviator). was this intended? if not, it could be evidence that the agents you model are "in a similar strategy space" than humans. in any case, it is remarkable.

What we may need to take into account: the suggested overall agent built with the no-press base machine and the communication implanted on top of this could actually limit the strategic space because they are separately designed (which is not so for a human player). I wonder if this is the case. Hard to tell, but it could be interesting if we can obtain something even stronger if both are sort of "evolved together". Don't know how to do that, though.

minor things I recommend to change:

early on page 2:

I think it would be good to add here that in a real game, player pairs usually separate for the negotiation, it is secret to the others.

Also maybe add something on the role of supply centers and armies? It comes in the conclusions but could as well put here already if needed anyways.

figure 1:

from the picture alone, it does not get clear at all what the role of the neural network is. the negotiation effect is pretty clear, the map is a bit small, also the text under the map. one may think of putting the player avatars directly into a larger map?

early in 2.2.3:

equal probabilities for Binary Negotiators or Sanctioning Agents? both in one game always or only one type?

btw, it is of high importance to know when in the game the deviation takes place, because time for sanctioning may be too short then

discussion 3:

(bottom of page 9, statement on DL approaches)

this is probably true, but in fact we do not really know as there is not such a vast body of recent Diplomacy AI research. DL approaches have taken it up and they look good, which is however no proof that other methods could not.

in 4.1.1:

where does the state value function come from, even if it is the same as for the non-press original agent, has it been learned from human data? or what does it contain?

in the supplemental material, concerning appendix 2:

how have these parameters been achieved?

Reviewer #3 (Remarks to the Author):

What are the noteworthy results?

This is a very interesting research where the authors introduce novel methods for AI agents to negotiate in complex environments.

Overall, these results extend and generalise the authors' own work in [43] to make possible that AI agents cope with the complexity that humans face when playing Diplomacy and still achieve cooperation. This is one of the things to praise in this work: the authors move forward from the simplifying assumptions made in Non-Press Diplomacy, and in [43], and investigate cooperation in a very realistic setting.

The authors augment their own agents employed for Non-Press Diplomacy in [43] to be capable of communicating and negotiating. This encompasses the design of two negotiation protocols (the mutual proposal protocol and the propose-choose protocol) plus novel negotiation algorithms for agents to reach agreements in both protocols. Again, I believe that the Propose-Choose protocol is very interesting because it actually captures the complexity of actual negotiations between humans in Diplomacy.

Another novelty is the sanctioning mechanisms that they endow agents with to retaliate deviators that do not honour their agreements. This is motivated by a real-life assumption: people may agree on a contract but then fail to meet its terms.

The authors show that augmented (negotiating) agents beat non-augmented (non-communicating) agents, and hence negotiation is beneficial. Therefore, there are significant benefits for agents capable of communicating and negotiating to achieve cooperation.

Furthermore, the authors devise a novel sanctioning mechanism that highly deters deviating agents, even in their more strategic version (learned deviators). Since even learned deviators highly honour their agreements, the results indicate that deviating from agreements is almost negligible thanks to sanctioning.

I also appreciate that the authors identify the weaknesses of their approach by conducting a thorough stability analysis of cooperation in the supplementary material.

Will the work be of significance to the field and related fields? How does it compare to the established literature? If the work is not original, please provide relevant references.

Overall, the paper tackles a very pertinent, open problem and presents significant contributions that advance the state of the art on cooperative AI.

Former work focuses on a simplified version of Diplomacy (No-press Diplomacy), a version of the game where communication and negotiation are not allowed.

This work takes a step ahead and focuses on the standard Press Diplomacy, a more complex version than Non-Press Diplomacy where communication between players is allowed.

Therefore, this work considers a problem with the same complexity faced by humans. In my opinion, and considering the list of noteworthy results above, this work is significant with respect to the state of the art and makes original and relevant contributions to understanding and implementing cooperation mechanisms for AI agents.

I agree with the authors that besides sanctioning there is potential for reputation mechanisms, also used by humans, to help AI agents deter deviators. This is indicated by the authors as future work. Although the authors cite work on reputation, I believe that they miss relevant references in the literature that consider the use of reputation in the realm of cooperation (e.g. [a], but also subsequent work by Griffiths on rewiring), and also in the realm of dynamic coalition formation and cooperation (several works by Peleteiro such as [b], [c])

[a] N. Griffiths and M. Luck. Changing neighbours: improving tag-based cooperation. In Proceedings of the 9th International Conference on Autonomous Agents and Multiagent Systems: Volume 1, AAMAS '10, pages 249–256, 2010.

[b] Peleteiro, A., Burguillo, J. C., & Chong, S. Y. (2014, May). Exploring indirect reciprocity in complex networks using coalitions and rewiring. In AAMAS (Vol. 14, pp. 669-676).

[c] Peleteiro, A., Burguillo, J. C., Luck, M., Arcos, J. L., & Rodríguez-Aguilar, J. A. (2015). Using reputation and adaptive coalitions to support collaboration in competitive environments. Engineering applications of artificial intelligence, 45, 325-338.

To the best of my knowledge, these works are not cited neither in the main text nor in [19].

Does the work support the conclusions and claims, or is additional evidence needed?

The authors do an excellent job at supporting their conclusions and claims. However, I believe that addressing the issues below would improve the paper.

The authors claim that their negotiation algorithms (RSS and MBDS) can be generalised to other contract types besides those considered in the paper. However, it is unclear to this reviewer how this is possible since no sufficient details are provided to back up this claim in section 2.1.

In Section 4 in the supplementary material, I believe that clearly identifying a formal proof discussing the convergence of MBDS to the Nash Bargaining solution would improve clarity.

When analysing the learned deviator behaviour (section 6 in the supplementary material), how shall we semantically interpret the chosen settings for the thresholds? It seems that t_{ψ} is set very low so that deviators are extremely cautious when considering the remaining opponent strength. This clearly dominates the other threshold, as evidenced by the experiments in section 2.2.3, and I wonder what would happen as this parameter increases. I guess that the learned deviator's wins decrease as t_{ψ} increases. Perhaps I'm wrong here, but it would be good if the authors comment on this.

Are there any flaws in the data analysis, interpretation and conclusions? Do these prohibit publication or require revision?

I have not identified any flaws that prevent publication.

However, I would recommend improving the clarity of the data analysis as follows.

First, the explanation of the behaviour of conditional deviators in section 2.2.1 is not sufficiently clear.

Second, the authors should be precise about the parameter α employed by sanctioning agents when maximising the metric $V_i(s') - \alpha V_j(s')$. What is the domain and semantics of this parameter? I guess that it controls the degree of importance placed on lowering player p_j 's expected value, but it should be explained in the main text.

In section 2.2.3 I think that it would be better to make clear that no defense in figure 7 corresponds to Baseline Negotiators.

I found it rather difficult to understand the graphical representation in figure 8, and also figures 5 and 6 in the supplementary material. I wonder whether the authors could think of a more comprehensible graphical representation. I'm not saying that this is necessary, but I think that the above-mentioned figures are a bit difficult to understand.

Is the methodology sound? Does the work meet the expected standards in your field?

The methodology is sound.

The authors first compare negotiating agents vs non-negotiating agents. Once they show the benefits of negotiation, they address how to tackle deviators with increasing degrees of complexity with defensive strategies with increasing degrees of complexity. I think that the authors do well by confronting defensive agents with optimised learned deviators, which offer the most challenging, strategic deviating behaviour. In this way the authors manage to empirically identify the best defensive strategy.

In terms of presentation, I would recommend to move subsection 4.6 (Learned Deviator) into subsection 4.4, which I understand details all deviator algorithms.

I have some observations about the methods employed.

It remains unclear to me how the contracts handled by RSS (algorithm 2 in section 4.3.1) are generated, while this issue is clear for MBDS. Does RSS employ a similar approach to MBDS?

Is there an error regarding the scaling factor $\frac{q^{0_{i,j'}}}{q^{0_{j',j}}}$ on page 14 (section 4.3.2) of the supplementary material?

The work meets the expected standards in my field.

Is there enough detail provided in the methods for the work to be reproduced?

Yes, indeed. The authors provide the source code that they employed as well as the values of the hyperparameters used in the experiments.

Revised version of the paper:

“Artificial Intelligence for the Board Game of Diplomacy:
Communicating, Negotiating and Promoting Honesty”

Dear Reviewers,

Thank you for all your detailed and helpful comments on our paper - we highly appreciate the effort and the insightful discussion!

We are pleased to submit a revised version of our paper, addressing all your comments. We first outline the main changes, and respond to each point raised below.

We have improved the presentation of the work and the discussion of related work based on your comments. You have indeed identified very relevant papers, which we now cite and discuss (please be aware that there are restrictions on the length of the main paper - we took care to remain within the limit). Our discussion of complementary mechanisms such as reputation systems is now more complete.

Based on your comments, we have also broadened the discussion of the reasoning behind some agent algorithms and parameters, and expanded the investigation of their impact. In particular, we discussed some possible strategic similarities between agent and human gameplay, and expanded the discussion of the limitations of our approach, such as separating out the agent's action policy and the negotiation policy (rather than “co-evolving” these). To address your questions, we have added several experimental results regarding agent parameters and behavior: the impact of the sanctioning parameter α on winrates, the differences between Conditional and Learned Deviators in terms of the time of the first deviation, and the impact of an extended range of the t_{ψ} parameter (controlling the Learned Deviator's threshold on the remaining opponent strength).

We have of course clarified the presentation based on your comments. We have also corrected all typos and grammatical errors you have noted, and improved the images/figures based on your comments.

Please see a point by point response below.

—

Point by point response:

Reviewer #1 (Remarks to the Author):

> This paper is amazing. It takes Diplomacy as sample problem for teaching agents how to negotiate non-binding agreements (as common in Diplomacy), and when to keep them or not keep them according to the overall chances of winning the game. In many ways, things nicely fall into place here, and it seems in the rearview so logical and clear. Which is, I think, good, because it shows that something has been achieved. In many ways, the sequence of agents that is presented reminds me of the hide-and-seek paper. And there is certainly room for more in this direction, but not all in a single paper.

>> Added. Thank you! Indeed, there are similarities to domains such as the Hide-and-Seek paper where one team employs measures or a strategy to achieve its goal, then the other side employs counter-measures to overcome these, then the first side employs a strategy to counter the counter-measures and so on. We have added a note on this issue in the discussion (and of course, we cite the Hide-And-Seek paper).

> This is definitively a big step (whatever limited in means of communication) into the direction of seeing flexible communication mechanisms evolved in agents. Actually, the next steps then may be to make the "characters" even more flexible (more types allowed in Figure 5), and eventually mix them in the sense that, as humans do, the agents can adapt their strategies to the environment. Which is of course more difficult to sort out and investigate than the defined single types. This direction (or whatever the authors think should be the next steps) may be made a bit more clearer in the intro already.

>> Done. Indeed, as you say, we view our paper as a step towards enabling flexible communication strategies to be evolved. As you suggest, we have added a part early in the introduction to emphasize this, and now write (see end of the Introduction):

"... we view our results as a step towards evolving flexible communication mechanisms in artificial agents, and enabling agents to mix and adapt their strategies to their environment and peers. "

> The most stunning surprise I had when reading the paper was that the strategies of the successful agents in the end very closely resemble successful human strategies. That is also an aspect that may be referred to more in the paper. Apart from the fact that it is quite rare that single players win the game when humans play, it is well known that a "buildup trust, deviate once, when it really leads to winning the game quickly" strategy is quite usual for human players. Which is very similar to the latest agents you present. It is very interesting to see that albeit partly different conditions (e.g. the amount of negotiations humans can actually do in the usually fixed time frame for negotiations is fairly limited), still very similar successful strategies "evolve". Did you drive it that way? Or did it just come out of how the first agents were "complexified"?

>> Added. Indeed, as you point out, the Learned Deviator adapts to follow through on its promises for a long period of the game (which can be viewed as building up trust, as you say), and then deviate when it leads to conclusively winning the game. We considered the Learned Deviator in the context of Sanctioning Agents who respond to deviations from agreements by actively attempting to lower the deviator's utility (holding a "grudge" until the end of the game). Hence, it makes sense to parametrize the choice of whether to deviate from an agreement on both the *gain from the deviation* and the *opponent's power to sanction the deviator*.

Before running the experiments, we did not know which behavior would emerge for the learned deviator; for instance, the results could have been dominated by the gains from deviation, or the deviator could have learned to deviate quite early in the game. We believe that the behavior that emerges for the Learned Deviator is a consequence of the nature of the Diplomacy game. Diplomacy is famous for having communication and trust as key elements of the game: "communication and trust are highly important; players must forge alliances with others and observe their actions to evaluate their trustworthiness. At the same time, they must convince others of their own trustworthiness while making plans to turn against their allies when least expected. A well-timed betrayal can be just as profitable as an enduring, reliable alliance." (from the Wikipedia entry: [https://en.wikipedia.org/wiki/Diplomacy_\(game\)](https://en.wikipedia.org/wiki/Diplomacy_(game))). Since Diplomacy is designed such that deviating from agreements can be very beneficial towards winning the game, so long as the opponent is left incapable of retaliating, it makes sense that agents can learn such a behavior given a rich enough representation of the Diplomacy game, along with communication and strategies in it.

To emphasize these points, we have added the following notes to section 3 (Discussion):

"... Humans playing Diplomacy sometimes employ a strategy of keeping promises until a crucial deviation late in the game [Niculae, 2015]. It's striking that our Learned Deviators follow a similar strategy: they adapt to typically follow through on their promises for long periods of time, breaking agreements only when the gains are high enough and when their opponents are left less capable of retaliating for the deviation. "

Please also see further note below regarding your comment on the "strategy space" that humans and the Learned Deviator face in Diplomacy.

> My expertise is more in the game AI direction but also in Diplomacy as a human player, not so much in general computational negotiation mechanisms, it may thus be that there is more from that field that shall be cited. But otherwise, references look good to me.

>> Added. Indeed, we agree that a more detailed discussion on computational negotiation mechanisms is warranted. We have added a discussion of this in the Supplementary Material, with several key surveys and papers - see Appendix 7, now titled "Appendix: Relation to Work on Computational Negotiation Mechanisms and the Evolution of Cooperation".

> There are some more papers on computational Diplomacy, I sense they are hard to find because "diplomacy" is not a good search term. Two that come to mind you may find interesting (up to you) are from this team (I have been evolved in these, so it is totally up to your choice if you want to take them into account):

Markus Kemmerling, Niels Ackermann, Nicola Beume, Mike Preuss, Sebastian Uellenbeck, Wolfgang Walz: Is human-like and well playing contradictory for Diplomacy bots? CIG 2009: 209-216. The authors find that if bots play too good (restricted press), they may be seen as too far from human play.

Markus Kemmerling, Niels Ackermann, Mike Preuss: Making Diplomacy Bots Individual. Believable Bots 2012: 265-288. This is somewhat similar, but finds that the expectations of humans how to actually identify bots compared to players are very different. Maybe not that surprising, but effectively that means that bots who may be seen as bots by one human are not recognized by another player because she/he has different expectations.

>> Added. These are indeed very relevant - thank you! We now cite these papers in the Discussion section (see the middle of page 10). We feel that given these, our results showing some similarities between human and agents strategies (e.g. following through on many agreements before a critical deviation from agreements) is noteworthy - these earlier papers note that strong performance may require behavior that is not "human-like" (and that different humans may have different expectations from artificial agents). See further notes below regarding the "strategy space".

> Overall, observations concerning the "strategy space" may be interesting. As already stated above, it looks as through subsequent experimentation with different strategies, a human-like playing style has evolved that seems superior at least on average (learned deviator). was this intended? if not, it could be evidence that the agents you model are "in a similar strategy space" than humans. in any case, it is remarkable.

>> Added. Indeed, our interpretation is that one of the features of Diplomacy as a game is that respecting agreed deals for many turns before a critical deviation can be an appealing high-level strategy. Of course, Full-Press Diplomacy is famous for dilemmas regarding trust. We feel like this property of the game is retained even under our simple restricted-press protocols. This could lead to the similarity between effective human gameplay and the behavior that emerges for the Learned-Deviator. We have added a note on this in the Discussion section (following the other comments we have added regarding the Learned Deviator and its possible similarity to human strategy); see note regarding the strategy space in the middle of page 10:

"We strived to create a rich representation of the Diplomacy game, along its dilemmas relating to strategic communication. One possible cause for the strategic similarity is that Diplomacy was designed such that deviating from agreements can be beneficial when the opponent would find it difficult to respond. In other words, Diplomacy, even under the restricted communication protocols we considered, has a "strategy space" where such a behavior can be an effective way to win, so both humans and our Learned Deviator exhibit it. "

> What we may need to take into account: the suggested overall agent built with the no-press base machine and the communication implanted on top of this could actually limit the strategic space because they are separately designed (which is not so for a human player). I wonder if this is the case. Hard to tell, but it could be interesting if we can obtain something even stronger if both are sort of "evolved together". Don't know how to do that, though.

>> Added. Absolutely, this is a limitation of the way we have built our Press Diplomacy agents. As you write, we take an agent trained in the No-Press version of the game, and infuse it with negotiation protocols and algorithms, rather than "co-evolving" these two parts. We now clearly discuss this limitation, and indeed we believe future work could devise stronger agents by improving this design aspect. We now write:

"...one limitation of our methods is that we take initial agents trained for No-Press Diplomacy, then augment them with negotiation algorithms. This means that the action strategy, coming from the base agent, was designed separately from the communication and negotiation strategies. Future work might build stronger agents through better methods for co-evolving the action and communication strategies in a way that is more similar to human play." (see the end of the conclusion).

> minor things I recommend to change:

>> We address each of these items separately.

> Early on page 2: I think it would be good to add here that in a real game, player pairs usually separate for the negotiation, it is secret to the others.

>> Done. Added a note that "players may converse privately by stepping into another room, or through a private chat channel when playing online"

> Also maybe add something on the role of supply centers and armies? It comes in the conclusions but could as well put here already if needed anyways.

>> Done. The intro now reads: "...an introduction to the rules of Diplomacy is given in Appendix A. Diplomacy is played on a map of Europe partitioned into provinces, some of which are special and marked as supply centers. Each player attempts to own the majority of the supply center, and controls multiple units (armies or fleets). A unit may support another unit (owned by the same or another player), allowing it to overcome resistance by other units.". We have also extended the game description in the supplementary material to emphasize the role of supply centers and armies and fleets.

> figure 1: from the picture alone, it does not get clear at all what the role of the neural network is. the negotiation effect is pretty clear, the map is a bit small, also the text under the map. one may think of putting the player avatars directly into a larger map?

>> Done. We have removed the neural network image, and made the map and text under it bigger (shifting the avatars to make room). Thank you for this comment - this has made the figure more readable.

> early in 2.2.3: equal probabilities for Binary Negotiators or Sanctioning Agents? both in one game always or only one type? btw, it is of high importance to know when in the game the deviation takes place, because time for sanctioning may be too short then.

>> Added. Thanks, we should have been clearer about this. Each of these games has agents of exactly two types: k Conditional Deviators playing against 7-k Binary Negotiators, or k Conditional Deviators playing against 7-k Sanctioning agents etc. We have added a note clarifying this in the caption of the figure.

Indeed, it is important to examine when a deviation takes place (which affects the amount of turns in which sanctioning occurs). The Deviators in these games are Conditional Deviators, who deviate at the first turn where they expect to gain a higher utility by not keeping the deal (i.e. their computed utility implicitly assumes that the other side does not modify their behavior). Hence, for Conditional Deviators, deviators typically occur early in the game (meaning that the sanctions, if evoked, are held in force for many turns). We have added a figure to Appendix H in the Supplementary Material showing the Number of contract-turns before the first turn in which a contract is broken, in games with a *Conditional Deviator* playing against Sanctioning Agents (this figure is similar to the figure just before it which considers the Learned Deviator, which allows contrasting the behavior of these two Deviator types), and a discussion of its implications.

> discussion 3: (bottom of page 9, statement on DL approaches). this is probably true, but in fact we do not really know as there is not such a vast body of recent Diplomacy AI research. DL approaches have taken it up and they look good, which is however no proof that other methods could not.

>> Fixed. Indeed, we wholeheartedly agree that other approaches, not based on deep learning, can also be used to successfully tackle the large action space of Diplomacy. We have rephrased to clarify this. The text now reads: "... Diplomacy makes an exceptional testbed: it has simple rules but high emergent complexity, and an immense action space which has recently been tackled using deep learning (future work could of course uncover other successful approaches)."

> in 4.1.1: where does the state value function come from, even if it is the same as for the non-press original agent, has it been learned from human data? or what does it contain?

>> Added. The techniques we apply take a state value function V as input. In our experiments, we take a value function from the base RL agent, trained to play No-Press Diplomacy (the one constructed in the NeurIPS paper on RL for Diplomacy: Anthony, Thomas, et al., Learning to play no-press diplomacy with best response policy iteration, NeurIPS 2020). That work starts by applying supervised learning to imitate human gameplay on a large dataset of Diplomacy games, then improves the agent through reinforcement learning, yielding a value function that is thus the product of both learning from human data and reinforcement learning improvements.

We have added a note clarifying this in Section 4.1 (bottom of page 10): "Various methods were proposed for constructing policy and value functions in Diplomacy. Our experiments use policy and value networks trained by first imitating human gameplay on a dataset of human Diplomacy games, then applying reinforcement learning to improve agent policies [Anthony, Thomas, et al] ([available at https://github.com/deepmind/diplomacy](https://github.com/deepmind/diplomacy)). Hence, the value and policy functions we use are based on both learning from human data and applying reinforcement learning improvements, but our algorithms can work with any such functions. "

> in the supplemental material, concerning appendix 2: how have these parameters been achieved?

>> Added. We generally selected hyperparameter values based on very coarse sweeps (for many hyperparameters the behavior remains similar across a relatively broad range of values). Some key hyperparameters, such as the number M of action profiles, and the number N of candidate action, relate to the *compute resources* spent on choosing each action (i.e. the runtime grows linearly as each of these is increased). In this case we have opted for as high values as we could afford, while still allowing us to run the experiments in a reasonable time (although increasing these values lead to better Monte-Carlo estimates, it causes a significant slowdown in the time it takes to run experiments). We have added a note on this (page 3 in the Appendix), saying that: "... The experiments we run are quite computationally demanding, and increasing these values would have slowed experiments down (though likely higher values result in more accurate estimates and hence possibly stronger agents)".

Reviewer #3 (Remarks to the Author):

> What are the noteworthy results?

This is a very interesting research where the authors introduce novel methods for AI agents to negotiate in complex environments.

Overall, these results extend and generalise the authors' own work in [43] to make possible that AI agents cope with the complexity that humans face when playing Diplomacy and still achieve cooperation. This is one of the things to praise in this work: the authors move forward from the simplifying assumptions made in Non-Press Diplomacy, and in [43], and investigate cooperation in a very realistic setting.

The authors augment their own agents employed for Non-Press Diplomacy in [43] to be capable of communicating and negotiating. This encompasses the design of two negotiation protocols (the mutual proposal protocol and the propose-choose protocol) plus novel negotiation algorithms for agents to reach agreements in both protocols. Again, I believe that the Propose-Choose protocol is very interesting because it actually captures the complexity of actual negotiations between humans in Diplomacy.

Another novelty is the sanctioning mechanisms that they endow agents with to retaliate deviators that do not honour their agreements. This is motivated by a real-life assumption: people may agree on a contract but then fail to meet its terms.

The authors show that augmented (negotiating) agents beat non-augmented (non-communicating) agents, and hence negotiation is beneficial. Therefore, there are significant benefits for agents capable of communicating and negotiating to achieve cooperation.

Furthermore, the authors devise a novel sanctioning mechanism that highly deters deviating agents, even in their more strategic version (learned deviators). Since even learned deviators highly honour their agreements, the results indicate that deviating from agreements is almost negligible thanks to sanctioning.

I also appreciate that the authors identify the weaknesses of their approach by conducting a thorough stability analysis of cooperation in the supplementary material.

>> Thank you! Absolutely, our focus was on showing how simple and natural communication and negotiation mechanisms can allow agents to jointly improve their welfare, and explore methods to encourage cooperation even when agreements are non-binding. As you write, the stability analysis of cooperation certainly indicates that the proposed mechanisms have some limitations. We have tried to fully discuss these, and hope the future work could identify techniques to further strengthen multiagent cooperation.

> Will the work be of significance to the field and related fields? How does it compare to the established literature? If the work is not original, please provide relevant references.

Overall, the paper tackles a very pertinent, open problem and presents significant contributions that advance the state of the art on cooperative AI.

Former work focuses on a simplified version of Diplomacy (No-press Diplomacy), a version of the game where communication and negotiation are not allowed.

This work takes a step ahead and focuses on the standard Press Diplomacy, a more complex version than Non-Press Diplomacy where communication between players is allowed. Therefore, this work considers a problem with the same complexity faced by humans. In my opinion, and considering the list of noteworthy results above, this work is significant with respect to the state of the art and makes original and relevant contributions to understanding and implementing cooperation mechanisms for AI agents.

I agree with the authors that besides sanctioning there is potential for reputation mechanisms, also used by humans, to help AI agents deter deviators. This is indicated by the authors as future work.

>> Thank you! Indeed, we tried to point out how future work on additional mechanisms, such as reputation systems, can further help agents cooperate with one another.

> Although the authors cite work on reputation, I believe that they miss relevant references in the literature that consider the use of reputation in the realm of cooperation (e.g. [a], but also subsequent work by Griffiths on rewiring), and also in the realm of dynamic coalition formation and cooperation (several works by Peleteiro such as [b], [c])

[a] N. Griffiths and M. Luck. Changing neighbours: improving tag-based cooperation. In Proceedings of the 9th International Conference on Autonomous Agents and Multiagent Systems: Volume 1, AAMAS '10, pages 249–256, 2010.

[b] Peleteiro, A., Burguillo, J. C., & Chong, S. Y. (2014, May). Exploring indirect reciprocity in complex networks using coalitions and rewiring. In AAMAS (Vol. 14, pp. 669-676).

[c] Peleteiro, A., Burguillo, J. C., Luck, M., Arcos, J. L., & Rodríguez-Aguilar, J. A. (2015). Using reputation and adaptive coalitions to support collaboration in competitive environments. Engineering applications of artificial intelligence, 45, 325-338.

To the best of my knowledge, these works are not cited neither in the main text nor in [19].

>> Added. Thank you for these references - they are indeed very relevant. We have of course added these citations, and a discussion on these where we consider reputation and cooperation and dynamic coalition formation (see the note towards the bottom of page 10).

> Does the work support the conclusions and claims, or is additional evidence needed?

The authors do an excellent job at supporting their conclusions and claims. However, I believe that addressing the issues below would improve the paper.

>> We address these, point by point below.

> The authors claim that their negotiation algorithms (RSS and MBDS) can be generalised to other contract types besides those considered in the paper. However, it is unclear to this reviewer how this is possible since no sufficient details are provided to back up this claim in section 2.1.

>> Done. We have added a discussion on this Appendices E and F in the Supplementary Material (and a reference to this in the Section 2.1 in the main text). Appendix E is now titled “Comparison and Extensions of our Negotiation Methods for Diplomacy”, and Appendix F is titled “Generalizing our Negotiation Methods Beyond Diplomacy”. These discuss how the protocols and algorithms could generalize to other settings. We now give examples of other contract types and show how the protocol/algorithm can work with these (beyond mutual Peace), and give some examples of how the methods can work with other environments.

One extension is examining other types of allowed contracts. We’ve used the mutual Peace contract for the Mutual Proposal protocol. This is just one contract type of many possible. Each such “type” of contract is a rule that examines the board state, and outputs which actions are allowed for each side, and which actions are disallowed. Under mutual Peace, neither side may move a unit to a province where there is a unit owned by the other player (or assist a third party in doing so). However, one could consider other such rules. For instance, a rule may preclude either side from convoying any unit using its fleet (“No Convoy”) or preclude either side from assisting any unit of any third party. The RSS protocol only can work with any such contract type, and only requires a concrete definition of the nature of the contract in the form of a rule that takes a board state and outputs the legal actions of the sides. The topic of other contract types is discussed in Appendix E.

Another example is considering domains other than Diplomacy. One example is that of fleet management, where participants try to coordinate the choice of roads used by vehicles in their fleet in order to reduce congestion on roads. This domain is one where participants take simultaneous actions that affect one another, and is discussed in Appendix F.

Note that we do not aim to claim that the protocols are applicable to all domains — we fully acknowledge that our methods have important limitations, discussed in the Discussion section of the main paper and Appendix C; Rather, we wanted to point out some simple extensions of the mechanisms we've proposed, that may be useful to others.

> In Section 4 in the supplementary material, I believe that clearly identifying a formal proof discussing the convergence of MBDS to the Nash Bargaining solution would improve clarity.

>> Done. We have cleaned up the section to clearly identify a formal proof. We have broken the original proof into multiple parts, which are now considered in more detail (see the revised Appendix D).

The principles behind the proof are simple. First, we observe that there is a finite (though large) set of actions each side may take, and as a result there is a finite set of possible contracts. MBDS creates a set of contracts to consider by sampling actions for the sides, so when the parameter determining the number of such samples is high enough, all contracts would be considered. Second, MBDS estimates the expected value of a contract for both sides by performing many simulations of how the next timestep would look like, and averaging the values of these 'simulated futures'. Similarly, it estimates the value of actions as the "no-deal" baseline by simulating possible future timesteps when the sides take these actions (and the other agents sample from their unrestricted policies). In both cases, these are i.i.d random variables whose expected values is the target quantity MBDS is estimating. Hence, by the law of large numbers, when there are infinitely many samples, the sample average should be arbitrarily close to the target quantity. We use Hoeffding's inequality to bound the probability of making a large error. Using the Hoeffding bounds, we show that by increasing the MBDS parameters we get MBDS to estimate the true score of each contract in the set of considered contracts with an arbitrarily high accuracy (hence, selecting the NBS contract, of true highest score).

We've formalized the proof, breaking it down to multiple parts. First we show that when the parameters of MBDS are set high enough, it selects the right "no-deal" baseline actions with high probability. We then show that by increasing the MBDS parameters, we can: a) make sure all contracts, and in particular the NBS contract, are in the set of considered contracts, and b) make the accuracy of the contract score estimates as high as desired. Combining these shows that the NBS contract is the one selected by MBDS when the parameters are high enough.

> When analysing the learned deviator behaviour (section 6 in the supplementary material), how shall we semantically interpret the chosen settings for the thresholds? It seems that t_{ψ} is set very low so that deviators are extremely cautious when considering the remaining opponent strength. This clearly dominates the other threshold, as evidenced by the experiments in section 2.2.3, and I wonder what would happen as this parameter increases. I guess that the learned deviator's wins decrease as t_{ψ} . Perhaps I'm wrong here, but it would be good if the authors comment on this.

>> Done. The parameter t_{ϕ} relates to minimum immediate improvement in value required to consider a deviation, and t_{ψ} relates to the maximum remaining opponent strength to allow a deviation. As you point out, the settings that the Learned Deviator finds are $t_{\psi}=0.03$ and $t_{\phi}=0.15$. Further, even a relatively small changes of ± 0.05 in t_{ψ} have a high impact on performance, whereas a change of ± 0.05 on t_{ϕ} typically has less impact in most settings. Intuitively, this means that: a) our Learned Deviator only deviates when the opponent would remain quite weak after the deviation, and b) empirically the Learned Deviator's performance is more sensitive to changes in the maximal allowed remaining opponent strength (t_{ψ}) than the immediate value gain parameter (t_{ϕ}). Based on your comment, we have run additional experiments with much larger values for t_{ψ} . As you predicted, when the t_{ψ}

parameter increases to significantly larger values, the winrate of the Learned Deviator drops (and for the much higher values, e.g. $t_psi > 0.2$, the Deviator underperforms with respect to the Sanctioning Agents). The top performing Learned Deviator params are $t_phi = 0.15$ and $t_psi = 0.03$. Fixing t_phi to that same value of $t_phi = 0.15$, we get the following winrates for various values of t_psi :

t_psi	Learned Deviator Winrate
0.01	1.009
0.03	1.017
0.08	1.012
0.2	0.997
0.35	0.991
0.5	0.984

We have added a note with this discussion to Appendix H (see page 14 of the Supplementary Material). Early in Appendix H we have also added a figure with the proportion of games where a deviation occurs, based on the two parameters t_phi and t_psi . This heatmap makes it easy to see the impact that these have on the Learned Deviator's behavior.

> Are there any flaws in the data analysis, interpretation and conclusions? Do these prohibit publication or require revision?

 I have not identified any flaws that prevent publication. However, I would recommend improving the clarity of the data analysis as follows.

>> We address these point by point

> First, the explanation of the behaviour of conditional deviators in section 2.2.1 is not sufficiently clear.

>> Done. We have cleaned up the explanation of the conditional deviator in section 2.2.1. In the previous version, we had a very brief description of the Conditional Deviator here in Section 2.2.1, and a full description and algorithms in Section 4.4. However, the description was indeed too short to be helpful, so as you suggest we have expanded this description, moving some content of Section 4.4 earlier to Section 2.2.1.

The description now reads:

“Similarly to Simple Deviators, when no deal has been agreed, a Conditional Deviator p_i selects actions from the unconstrained policy. However, when a deal $D = (R_i, R_j)$ is agreed on with another player p_j , the Conditional Deviator considers multiple actions it might take (these are sampled from its policy, and include candidate actions $a \notin r_i$ that violate the contract). For each such action, it performs multiple simulations of how the next turn might look like, by sampling actions for p_j from the policy constrained by the contract, reflecting the assumption that p_j would honor the agreement and thus refrain from taking an action $a \notin R_j$. The Conditional Deviator uses its average value in the sample to estimate the expected next turn utility for each of the candidate actions, and selects the action that maximizes its expected value in these simulations. The Deviator agents are fully described in Section 4.4...”

> Second, the authors should be precise about the parameter α employed by sanctioning agents when maximising the metric $V_i(s') - \alpha V_j(s')$. What is the domain and semantics of this parameter? I guess that it controls the degree of importance placed on lowering player p_j 's expected value, but it should be explained in the main text.

>> Done. Indeed, we should have been more detailed here. As you wrote, the parameter $\alpha \geq 0$ indeed controls the importance placed on lowering the deviator's expected utility (probability of winning the game). A value of $\alpha = 0$ reflects the Baseline Negotiator behavior of not sanctioning at all, whereas a value of $\alpha = 1$ reflects equal importance on increasing the Sanctioning Agent's utility and on lowering the Deviator's utility. As α approaches infinity, the Sanctioning Agent tends to focus on making sure the Deviator loses the game (with no regard to its own probability of winning).

The text in Section 2.2.2. now reads: "... a Sanctioning Agent considers both its own value $V_i(s')$ and the deviator's value $V_j(s')$ to rank actions. It maximizes the metric $V_i(s') - \alpha V_j(s')$ which consists of both improving its own probability of winning $V_i(s')$ and lowering p_j 's probability of winning $V_j(s')$. The parameter $\alpha \geq 0$ controls the relative importance placed on the two goals, with a value of $\alpha = 0$ reflecting no sanctioning behavior (equivalent to Baseline Negotiators) and $\alpha = 1$ reflecting equal importance on lowering the Deviator's utility. As $\alpha \rightarrow \infty$, the Sanctioning agent focuses solely on making the deviator lose the game (without regard to its own odds of winning the game). Note that Sanctioning Agents themselves do not deviate from their contracts."

> In section 2.2.3 I think that it would be better to make clear that no defense in figure 7 corresponds to Baseline Negotiators.

>> Done. We changed the "No Defense" label to directly refer to the Baseline Negotiator (i.e. it says Baseline rather than "No Defense").

> I found it rather difficult to understand the graphical representation in figure 8, and also figures 5 and 6 in the supplementary material. I wonder whether the authors could think of a more comprehensible graphical representation. I'm not saying that this is necessary, but I think that the above-mentioned figures are a bit difficult to understand.

>> Added. Indeed, these figures convey a lot of information. Figure 8 captures the impact of the Learned Deviator's two parameters on its winrate against Sanctioning Agents. The figure is a "heat map", where the x location of a cell refers to the value of the first parameter and the y location of a cell refers to the second parameter. Hence, each cell in the figure relates to a specific parameter setting (specific values for the two parameters of the Learned Deviator), and the color of the cell represents the Learned Deviator's winrate under this setting.

Such heatmaps are fairly standard figures, but we wanted to also include data regarding the variance in the Learned Deviator's performance under these settings. The color of each cell is computed by running many games with a Learned Deviator (with the parameter values that the cell relates to) against Sanctioning agents. Due to stochasticity in the agent policies, each such game has a different outcome, and the color of the cell describes the average winrate in these games, i.e. the proportion of these games that the Learned Deviator won. The variance in whether the agent won or lost in these games is important, as it allows us to build a confidence interval that contains the true winrate with high probability. For example, we would like to express the fact that our approximated winrate is $80\% \pm 10\%$ (i.e. with high probability the true winrate is between 70% and 90%). We describe the variance using the two circles in the cell (the high bound and low bound of the confidence interval). This is similar to how error bars are used in other figures, except here we have a 2D rather than 1D surface.

We acknowledge that the figure is somewhat harder to understand, and have experimented with alternatives. One possibility is simply eliminating the small circles in each cell (i.e. make the figure a standard heatmap), but this loses information regarding the confidence intervals. Another alternative is a contour map rather than a heatmap, but this was even harder to read.

We have opted to keep the figures in the same format, and instead add a clarification regarding the roles of the circles in the cell. The text reads: "... Impact of the Learned Deviator parameters t_ϕ, t_ψ on the winrate advantage when playing against a population of 6 Sanctioning agents. The plot shows the ratio $\frac{W_{Dev}}{W_{San}}$ between the Deviator's winrate W_{Dev} and the Sanctioning Agent's winrate W_{San} . The x-axis location of a cell relate to the parameter t_ψ and the y-axis location of a cell relates to the parameter t_ϕ . The color of each cell is the mean winrate of the Learned Deviator of these parameters (blue indicates the Learned Deviator outperforms the Sanctioning agents, and red indicates it under-performs). The colors of the top and bottom circles in each cell indicate a 95% confidence interval bounds for the winrate, given the sampled game (to be interpreted similarly to error bars)..."

> Is the methodology sound? Does the work meet the expected standards in your field?

The methodology is sound. The authors first compare negotiating agents vs non-negotiating agents. Once they show the benefits of negotiation, they address how to tackle deviators with increasing degrees of complexity with defensive strategies with increasing degrees of complexity. I think that the authors do well by confronting defensive agents with optimised learned deviators, which offer the most challenging, strategic deviating behaviour. In this way the authors manage to empirically identify the best defensive strategy.

> In terms of presentation, I would recommend to move subsection 4.6 (Learned Deviator) into subsection 4.4, which I understand details all deviator algorithms.

>> Done. This is indeed a much better flow for the methods section as all Deviator algorithms are considered together.

> It remains unclear to me how the contracts handled by RSS (algorithm 2 in section 4.3.1) are generated, while this issue is clear for MBDS. Does RSS employ a similar approach to MBDS?

>> Added. The Mutual Proposal protocol considers a single "type" of contract, in contrast to the Propose-Accept protocol which allows for many diverse contracts. Under Propose-Accept, a contract lists the exact action every unit of both sides would take in the next turn, making the space of contracts very large. In contrast, under Mutual Proposal each side either makes a proposal to the other side, or refrains from making such a proposal. Hence, the Mutual Proposal protocol depends on having a definition of the exact nature of the proposal, which we call the "type" of contract used under the protocol. Each such "type" of contract is a rule that examines the board state, and outputs which actions are allowed for each side, and which actions are disallowed.

For the Mutual Proposal protocol, our empirical analysis is based on a type of contract called the "Mutual Peace" contract. Intuitively, this type of contract means that (once it is agreed) neither side may attack the other; More concretely, if Peace is agreed on, neither side may move any of its units to a province occupied by a unit of the other side or to a supply center owned by the other side, and may also not assist a unit of a third party in doing so. The Peace contract says the two sides are at peace and may not take any offensive action against the other, but what constitutes an offensive action depends on the board state. For instance, suppose Germany and France agree on Peace. If in the current board state there is a German unit sitting in Burgundy, France is precluded from moving one

of its units into Burgundy (e.g. if France has a unit in Paris, the Peace contract means France cannot move that unit to Burgundy). On the other hand, if there is no German unit sitting in Burgundy, then France is allowed to move its unit from Paris to Burgundy. In other words, Mutual Proposal has a contract that reads “The two sides are at mutual peace, and neither side may attack the other” (in contrast to the set of many contracts possible under the Propose-Accept protocol).

While the specific actions this contract precludes a player from doing depend on the board state, the type of contract is the same throughout the game. Hence, under the Mutual Proposal protocol, a player needs only indicate that they are interested in “a contract” with another player, without indicating the specific nature of the contract (which is implicitly specified to be mutual Peace). The message sent by an agent is either “I am interested in a contract (mutual Peace) with player p_j ” or “I am not interested in a contract (mutual Peace) with player p_j ”. As we discuss in Appendix E in the supplementary material, while our analysis is based on the Peace contract, one can use Mutual Proposal with other types of contracts (that specify other rules regarding which actions are allowed in each state). In the case of using an alternative contract, the protocol and messages exchanged between agents remain exactly the same, but the set of disallowed actions when a contract is agreed is different (in appendix E we give several examples of other contract types, such as a “No-Move” contract, or “No-Convoy” contract).

Hence, the only decision taken by the RSS method for Mutual Proposal is to decide whether to send the message indicating the desire to have a contract with a peer p_j or to refrain from sending it (equivalent to saying that the player is **not** interested in a contract with p_j). Therefore, the way in which contracts are generated by MBDS is different from how RSS handles this. RSS does not generate contracts, but rather assumes exactly one type of contract exists (and uses the rules set by that contract type to determine which actions are possible under the contract and which are not). In contrast, under Propose Accept there is a large space of possible contracts, and the MBDS algorithm includes a part that generates such possible contracts.

Based on your comment, we have expanded the discussion in Appendix E to discuss this (and it now includes some examples of alternative contract types). We have further cleaned up the discussion in the main text that relates to this (see changes in Section 2.1). We hope these changes help clarify this issue.

> Is there an error regarding the scaling factor $\frac{q^0_{i,j}}{q^0_{j,j}}$ on page 14 (section 4.3.2) of the supplementary material?

>> Fixed, thanks for catching the typo (arising from an older notation we have changed). The text regarding the meaning of this scaling factor was correct: this scaling factor is the ratio between the baseline utilities for i and for j when no contract is agreed (i.e. when each selects an action optimizing solely for its own utility).

>The work meets the expected standards in my field.

> Is there enough detail provided in the methods for the work to be reproduced?

> Yes, indeed. The authors provide the source code that they employed as well as the values of the hyperparameters used in the experiments.

REVIEWERS' COMMENTS

Reviewer #1 (Remarks to the Author):

This revised version of the paper is ready for publication in my view. The many smaller, non-fundamental issues raised by the reviewers have all been fixed as far as I can see which definitively results in a stronger paper. I have no further requests on how to improve it.

Reviewer #3 (Remarks to the Author):

I would like to thank the authors for the effort that they have put in this revised version. I very much appreciate that the authors have carefully and satisfactorily addressed all my comments. I particularly appreciate the effort they have put on producing Appendix D, which now is excellent. In my opinion, the paper has largely improved and deserves publication.

Reviewer #4 (Remarks to the Author):

I find this diplomacy sandbox interesting, and the questions a light how to deter deception and deal breaking in repeated interactions to be of interest, along with the added twist about when the players are ai agents.

With that said, this is somewhat outside of my area of expertise (which is primarily human behavior, game theory, and evolutionary modeling). So I will keep my comments brief and happy to have them overridden.

But I am wondering why the diplomacy sandbox, and ai agents, are the best way to study the question of deception and deal breaking and not more standard game theory tools, like sub game perfection, and repeated games?

I also wasn't sure why in this particular setting, the agents that sanction deal breakers don't themselves just deviate and fail to sanction when expected to, given that sanctioning is costly--a classic issue raised in the evolution of cooperation literature.

Anyhow I find this intriguing. And perhaps there are obvious answers to my questions in which case, I look forward to learning more.

Revised version of the paper:

“Communication, Negotiation and Honesty in Artificial Intelligence Methods for the Board Game of Diplomacy”

Dear Reviewers,

Thank you for all your helpful comments on our paper - we appreciate the effort and the insightful discussion!

We are pleased to submit a revised version of our paper, addressing all your comments. This is the second revision, which mostly addresses two points. The first relates to more standard game theory tools, like sub game perfection, and repeated games, and the second relates to why agents that sanction deal breakers don't themselves deviate or fail to sanction when expected to, given that sanctioning is costly. We address both of these in this new revision. We have included a discussion of these issues in the revised Discussion section, and hope this would make the results accessible to the broader readership.

First, we discuss applying standard game theoretic tools, and in particular subgame perfect equilibria. We explain that such analysis can be very illuminating and allow constructing capable agents, but also the computational difficulties that make it hard to solve a game such as Diplomacy for a subgame perfect equilibrium. In Diplomacy, the large game tree size makes it intractable to directly calculate a subgame perfect equilibria (or refinements of a Nash equilibrium). We also highlight some key differences between Diplomacy and infinitely repeated games (such as Iterated Prisoner's Dilemma). Second, we extended the discussion of how sanctions ameliorate but do not completely solve the problem of deviations; In particular, we emphasize our results on the potential cost of sanctioning and the discussion of the case of agents who cease sanctioning deviations, and further discuss the need for additional mechanisms (such as repeating interaction across multiple games or a trust and reputation system) to fully deter deviations.

Please see a point by point response below.

—

Point by point response:

Reviewer #1 (Remarks to the Author):

> This revised version of the paper is ready for publication in my view. The many smaller, non-fundamental issues raised by the reviewers have all been fixed as far as I can see which definitively results in a stronger paper. I have no further requests on how to improve it.

>> Thank you again for your helpful comments!

Reviewer #3 (Remarks to the Author):

> I would like to thank the authors for the effort that they have put in this revised version. I very much appreciate that the authors have carefully and satisfactorily addressed all my comments. I particularly appreciate the effort they have put on producing Appendix D, which now is excellent.
In my opinion, the paper has largely improved and deserves publication

>> Thank you again for your helpful comments!

Reviewer #4

> I find this diplomacy sandbox interesting, and the questions a light how to deter deception and deal breaking in repeated interactions to be of interest, along with the added twist about when the players are AI agents.

>> Thank you for the comments. Indeed, we believe this is an interesting Sandbox regarding how to deter deceptions in a complex and large-scale negotiation domain.

> With that said, this is somewhat outside of my area of expertise (which is primarily human behavior, game theory, and evolutionary modeling). So I will keep my comments brief and happy to have them overridden.

> But I am wondering why the diplomacy sandbox, and AI agents, are the best way to study the question of deception and deal breaking and not more standard game theory tools, like sub game perfection, and repeated games?

>> Applying standard game theory tools, such as subgame perfect equilibrium and repeated games, is indeed a very important issue, which we now better address in the introduction and discussion. We have also added Appendix 8, titled: "On the Computational Difficulty of Solving the Game of Diplomacy". In short, we first note that such approaches (subgame perfect equilibria or repeated game analysis) are valuable tools for developing agents, but also note that while solving for a subgame perfect equilibrium is appealing in theory, doing so in practice is computationally hard (intractable) due to the large size of the game of Diplomacy.

We have added the following paragraph in the Discussion:

"... Diplomacy makes an exceptional testbed: it has simple rules but high emergent complexity, and an immense action space which has recently been tackled using deep learning ... Game theory offers powerful tools for analyzing games such as Diplomacy and for constructing agents. In particular, solving for a subgame perfect equilibrium or applying repeated game analysis could be useful for building capable agents. However, the large action space in Diplomacy results in an incredibly large game tree size, which impedes game theoretic analysis (see Appendix 8). We thus combine reinforcement learning methods with game theoretic solutions (the Nash Bargaining Solution), and address computational barriers using a Monte-Carlo simulation of potential future game trajectories. "

As we now explain in the paper:

"Diplomacy is interesting as it is an externally defined AI challenge and a game played by people (similarly to Chess, Backgammon or Stratego). Further, it has seven players and is thus richer than two-player zero-sum games, and can serve as a good example for a setting where players must form temporary alliances with some peers so as to overcome others. Another realistic feature of Diplomacy is the need to take multiple interrelated decisions simultaneously (in Diplomacy this relates to deciding how to move multiple units all at once). However, these exact properties that make Diplomacy interesting also pose a computational barrier for game theoretic analysis, which typically relies on solving games for a Nash equilibrium, or refinements such as a subgame perfect equilibrium."

We wholeheartedly agree that a subgame perfect equilibrium is a very valuable framework for studying what might occur in extensive form games. In theory, one could certainly solve the Diplomacy game to find a subgame perfect equilibrium (or possibly other refinements of a Nash equilibrium), and use this to predict game outcomes or guide agent behavior. However, the difficulty with achieving this in practice is a computational one.

We now further explain in the paper:

"Solving for a Nash equilibrium is computationally hard (see [Daskalakis, et al., the complexity of computing a Nash equilibrium]) making this a challenge even in small games. As we discuss in the introduction, Diplomacy is an enormous game. Selecting moves for multiple units at the same time results in a vast action space of 10^{21} to 10^{64}

legal actions per turn, and a rough estimate of the game tree size of Diplomacy is 10^{900} [Anthony et al.] ... (we contrast this with the game of Chess, which has fewer than 100 legal actions per turn, and a game tree size of 10^{123}). The sheer size of the game tree of Diplomacy and the computational complexity of applying game theoretic analysis means that in practice using such tools to build AI agents requires solving complex algorithmic challenges [Kraus et al., Anthony et al., Gray et al].

Similarly to recent work on smaller games such as Chess, Backgammon, or Go [Tesauro et al, Silver et al.] we chose to use a reinforcement learning based method to identify strong agent policies. Our approach combines reinforcement learning with game theoretic principles such as the Nash Bargaining Solution, applying a Monte-Carlo technique to overcome computational barriers. Such a combination of reinforcement learning and algorithmic game theory methods could fuel further successes in developing agents in the future.“

>> Specifically regarding game theoretic analysis of repeated games: Diplomacy is a temporally extended game, where players interact with one another over multiple turns, but there are also some differences from classical repeated games (where multiple players play a simple stage game repeatedly with one another), as in each turn the board configuration changes (i.e. in each turn the interaction is not exactly the same). There is some existing work on cooperation in small-scale repeated games, with some work on settings with incomplete information studying issues such as mixing strategies to reduce exploitability, strategically withholding information, misrepresenting intents or bluffing. We examined such issues and the conditions for honest cooperation in the large-scale temporally extended setting of Diplomacy, which we feel nicely complements the work on repeated games. Appendix 7, titled “Relation to Work on Computational Negotiation Mechanisms and the Evolution of Cooperation”, contains a comparison to repeated games, focusing on the settings studied in Axelrod’s Evolution of Cooperation and follow-up work. We have extended the notes there to better contrast the setting of Diplomacy and repeated games.

In Section 2, we now write:

“The full algorithms appear in Section 4 (Methods). Our Defensive Agents reproduce human behaviors of ceasing to trust peers who break promises or sanctioning such deviations, inspired by work on the Evolution of Cooperation [Axelrod] (see comparison in Appendix 7, as well as a discussion of differences between Diplomacy and repeated games). We examine the conditions for honest cooperation in the large-scale temporally extended setting of Diplomacy, complementing work on cooperation in repeated games, where multiple players repeatedly interact by playing an identical simple stage game with one another. “

Having a repeated interaction between the same players across multiple games of Diplomacy, or applying a reputation based system may yield more truthful collaboration between agents. In the revised Appendix 7 we now explain:

“Our design of the Defensive agents is influenced by Axelrod’s line of work on the Evolution of Cooperation [Axelrod and Hamilton]... The setting studied in the Evolution of Cooperation work and subsequent work is that of a repeated game, where participants play the same simple stage game such as Prisoner’s Dilemma repeatedly with the same partner. More generally repeated games, where multiple players play a simple stage game repeatedly with one another, were used to examine related topics such as mixing strategies to reduce exploitability, strategically withholding information, misrepresenting intents or bluffing.

Real-world negotiation has a number of attributes that differ from repeated stage games, and that Diplomacy captures more closely. Negotiating actors may be sometimes in conflict regarding some issues and interests, and may sometimes, even at the same time, be cooperating on some other issues. Actors may have a number of different negotiating partners to choose from, each with their own differing strengths, goals, and plans. Further, rather than taking turns to respond to each other’s moves, real-world negotiators are often managing multiple relationships in parallel.

More formally, our research complements work on repeated games and the Evolution of Cooperation as Diplomacy differs from repeated games and Iterated Prisoner’s Dilemma in multiple ways. First, we consider a single game that is temporally extended, rather than a repeated stage game. Hence, in a single Diplomacy game agents are extremely unlikely to meet under the exact same circumstances, but rather interact again in different board states, in which their

relative strength may be different. This means that the possible gains from deviation can vary widely across the game.

Furthermore, in our case the Shadow of the Future is the result of a change in the behavior of the defensive agent in subsequent turns of a single game, rather than multiple games. For infinitely repeated stage games there are prominent results in game theory stating that there exist fully cooperative Nash equilibria (Fudenberg and Maskin, Hofbauer and Sigmund), but as our setting relates to a single temporally extended game, such results do not apply (and indeed, establishing a fully cooperative equilibrium proves to be challenging).

For the setting of a single Diplomacy game, sanctioning deviation from contracts ameliorates the problem of strategic deviation from past agreements, but the Learned Deviator still outperforms full truthful Sanctioning peers. However, having players interact repeatedly in multiple Diplomacy games extends the interaction time horizon, resulting in a larger potential future loss upon a deviation, which could increase the incentive of agents not to break agreements. This could be a very interesting avenue for future research.

We further note that as Diplomacy has an incredibly large action space, agents may consider a very large space of behaviors. This is in contrast to only two strategies in Prisoner's Dilemma, "cooperating" or "defecting". In Diplomacy the undesired behavior, of agreeing to a contract and then selecting an action that violates the contract, might take on many forms, as there are many possible ways to deviate from an agreed contract.

Finally, the different modes of behavior change, ceasing to communicate for Binary Negotiators and active retaliation for Sanctioning agents, produce different effect sizes. Ceasing to communicate can be viewed as simply ceasing to trust that the other side would adhere to agreements, whereas sanctioning takes a more active role in lowering the deviator's utility. Iterated Prisoner's Dilemma cannot capture such different effects due to its very limited strategy space. "

> I also wasn't sure why in this particular setting, the agents that sanction deal breakers don't themselves just deviate and fail to sanction when expected to, given that sanctioning is costly--a classic issue raised in the evolution of cooperation literature.

>> You point out important questions regarding the Sanctioning agents: what happens when the Sanctioning agents allow themselves to deviate from their own agreements, and whether they could cease sanctioning deal-breakers as this sanctioning behavior is costly. These are indeed very valid questions, and as you point out, these are also classical issues raised in game theory. In the Discussion section of the previous revision we have pointed out that the sanctioning behavior is not an ironclad defense that can stop deviations altogether. Given your comment, in the revised Discussion section we added an extended note, focusing on these specific points and now write:

"Learned Deviators do gain a slight advantage against Defensive agents, and when many agents deviate from agreements sanctioning deviations becomes more costly. Hence, Sanctioning agents may themselves start deviating from agreements, and if the sanctioning behavior proves to be costly, the Sanctioning agents may cease sanctioning deviators. In Appendix 11 we probe the potential learning dynamics of an agent population with regards to sanctioning and deviation: we consider Sanctioning agents who themselves deviate from agreements, and the incentive to cease sanctioning peers when this sanctioning behavior is costly. We find that a population of sanctioning agents is only at a near-equilibrium (in contrast to simple repeated games such as Iterated Prisoner's Dilemma which may admit a fully stable cooperative equilibrium [Axelrod et al]).

Such issues have been studied in the literature on the evolution of cooperation, and for simpler games researchers noted that additional mechanisms might be required to support sustained cooperation, such as having repeated interaction. Similarly for Diplomacy, to avoid a population of learning agents gradually becoming less cooperative one might need to employ additional mechanisms. Repeating the interaction and playing multiple games of Diplomacy against the same opponents could increase the incentives for cooperative behavior. Further, we focused on sanctioning, but humans rely on diverse solutions to deter deviating from agreements such as employing trust and reputation systems [Sierra et al., Sabater et al.], or relying on a judicial system to provide remedies for the breach of

contracts [Farnsworth et al]. Future work could investigate how to implement such mechanisms for AI agents, similarly to earlier work on deterring deviations in dynamic coalition formation settings, for instance keeping track of peer behavior over multiple interactions or leveraging indirect reciprocity and tagging.”

Appendix 11 in the Supplementary Material (titled “Stability Analysis of Agent Cooperative Behavior”) considers the case of Sanctioning agents who themselves can deviate from agreements, and the incentives to cease sanctioning as this is costly. To quickly summarize the results discussed there, we note that the Learned Deviator who best responds to Sanctioning peers learns to very rarely deviate from agreements: in roughly 98% of the games no deviation occurs; Thus being a Sanctioning agent is “cost-free” in 98% of the games (no deviation occurs, so no sanctioning occurs either). Appendix 11 considers Sanctioners who themselves deviate from agreements, called Combined Agents (combining the behavior of the Sanctioning Agents with that of the Learned Deviator). We show that in a setting where all agents are Combined Agents deviations remain very rare and the incentive to cease sanctioning is very small (even after playing 10,000 games, it is difficult to show that this incentive is positive with high confidence). However, having agents cease sanctioning deviations could lead to a problematic cycle where deviations become more frequent, leading to a non-cooperative outcome.

We have thus included a note in this appendix as well, indicating that achieving cooperation could require additional mechanisms, such having repeated interaction across games, or using trust and reputation systems.